# Temporal Dynamic Quantization for Diffusion Models

**Junhyuk So**[1*]   **Jungwon Lee**[2*]   **Daehyun Ahn**[3]   **Hyungjun Kim**[3]   **Eunhyeok Park**[1,2]

[1] Department of Computer Science and Engineering, POSTECH
[2] Graduate School of Artificial and Intelligence, POSTECH
[3] SqueezeBits. Inc

## Abstract

The diffusion model has gained popularity in vision applications due to its remarkable generative performance and versatility. However, high storage and computation demands, resulting from the model size and iterative generation, hinder its use on mobile devices. Existing quantization techniques struggle to maintain performance even in 8-bit precision due to the diffusion model's unique property of temporal variation in activation. We introduce a novel quantization method that dynamically adjusts the quantization interval based on time step information, significantly improving output quality. Unlike conventional dynamic quantization techniques, our approach has no computational overhead during inference and is compatible with both post-training quantization (PTQ) and quantization-aware training (QAT). Our extensive experiments demonstrate substantial improvements in output quality with the quantized diffusion model across various datasets.

## 1   Introduction

Generative modeling is crucial in machine learning for applications such as image [1, 2, 3, 4, 5, 6, 7, 8], voice [9, 10], and text synthesis [11, 12, 13]. Diffusion models [3, 2, 4], which progressively refine input images through a denoising process involving hundreds of iterative inferences, have recently gained prominence due to their superior performance compared to alternatives like GANs [14]. However, the high cost of diffusion models presents a significant barrier to their widespread adoption. These models have large sizes, often reaching several gigabytes, and demand enormous computation of iterative inferences for a single image generation. Consequently, executing diffusion models on resource-limited mobile devices is practically infeasible, thus most applications are currently implemented on expensive, high-performance servers.

To fully exploit the potential of diffusion models, several methods to reduce the computational cost and memory requirement of diffusion models while preserving the generative performance have been proposed. For example, J. Song et al. [2] and L. Liu et al. [15] proposed a more efficient sampling scheduler, and T. Salimans & J. Ho proposed to reduce the number of sampling steps using knowledge distillation technique [16, 17]. As a result, high-fidelity images can be generated with fewer sampling steps, making the use of diffusion models more affordable and feasible. Despite these advancements, the denoising process of diffusion models still demands a substantial computational cost, necessitating further performance enhancements and model compression.

While the majority of previous approaches have focused on reducing the number of sampling steps to accelerate the denoising process, it is also important to lighten the individual denoising steps. Since the single denoising step can be regarded as a conventional deep learning model inference, various model compression techniques can be used. Quantization is a widely used compression technique where both weights and activations are mapped to the low-precision domain. While advanced quantization schemes have been extensively studied for conventional Convolution Neural Networks (CNNs) and

---

*Equal Contribution

37th Conference on Neural Information Processing Systems (NeurIPS 2023).

$$\textbf{Fowrad Process} \quad q(x_t|x_{t-1}) = N(x_t; \sqrt{1-\beta_t}x_{t-1}, \beta_t I)$$

$$x_0 \sim p(x) \qquad\qquad x_T \sim N(0, I)$$

$$\textbf{Reverse Process} \quad p_\theta(x_{t-1}|x_t) = N(x_{t-1}; \mu_\theta(x_t, t), \sigma_t^2 I)$$

Figure 1: The forward and reverse processes of diffusion models.

language models, their application to diffusion models has shown significant performance degradation. Due to the unique property of diffusion models, such as the significant changes in the activation distribution throughout the iterative inferences, the output is heavily distorted as the activation bit-width decreases [18]. Existing quantization techniques, including both quantization-aware training (QAT) [19, 20, 21] and Post-training quantization (PTQ) [22, 23, 24, 25], are designed to address a specific distribution in existing DNNs, therefore cannot deal with time-variant activation distributions in diffusion models.

To tackle the unique challenges of diffusion model quantization, we introduce a novel design called Temporal Dynamic Quantization (TDQ) module. The proposed TDQ module generates a time-dependent optimal quantization configuration that minimizes activation quantization errors. The strong benefit of the TDQ module is the seamless integration of the existing QAT and PTQ algorithms, where the TDQ module extends these algorithms to create a time-dependent optimal quantization configuration that minimizes activation quantization errors. Specifically, the module is designed to generate no additional computational overhead during inference, making it compatible with existing acceleration frameworks without requiring modifications. The TDQ module significantly improves quality over traditional quantization schemes by generating optimal quantization parameters at each time step while preserving the advantages of quantization.

## 2 Backgrounds and Related Works

### 2.1 Diffusion Model

Diffusion models have been first introduced in 2015 [26] and revolutionized image generation by characterizing it as a sequential denoising process. As shown in Fig. 1, the forward diffusion process gradually transforms the image($x_0$) into a random data($x_T$) which follows standard normal distribution by adding small Gaussian noise at each time step. The reverse diffusion process generates a clean image($x_0$) from a random data($x_T$) by gradually removing noise from the data through iterative denoising steps. Therefore, a diffusion model learns the reverse process which is estimating the noise amount on a given noisy data at each time step($x_t$). The forward($q$) and reverse($p_\theta$) processes can be described as

$$q(x_t|x_{t-1}) = N(x_t; \sqrt{1-\beta_t}x_{t-1}, \beta_t \mathbf{I}), \tag{1}$$

$$p_\theta(x_{t-1}|x_t) = N(x_{t-1}; \mu_\theta(x_t, t), \sigma_t^2 \mathbf{I}), \tag{2}$$

where $\beta_t$ denotes the magnitude of Gaussian noise.

[3] introduced a reparameterization trick for $\mu_\theta$ and the corresponding loss function, which facilitates the training of diffusion models.

$$\mu_\theta(x_t, t) = \frac{1}{\sqrt{\alpha_t}}(x_t - \frac{\beta_t}{\sqrt{1-\bar{\alpha}_t}}\epsilon_\theta(x_t, t)) \tag{3}$$

$$L_{simple} = E_{t,x_0,\epsilon}[||\epsilon - \epsilon_\theta(x_t, t)||] \tag{4}$$

While diffusion models can produce high-quality images, the iterative denoising process makes diffusion models difficult to be used for real-world scenarios. The early works on diffusion models such as DDPM [3] required hundreds to thousands of iterative inferences to generate single image, resulting extremely slow sampling speed. Therefore, numerous studies have been investigating algorithmic enhancements and various optimizations to boost performance, aiming to make diffusion

models more efficient and suitable for real-world applications. DDIM [2] introduced an implicit probabilistic model that reinterprets the Markov process of the DDPM method achieving competitive image quality with one-tenth of denoising stpes. Distillation-based methods [16, 17] proposed to reduce the number of denoising steps using knowledge distillation techniques.

On the other hand, to address the significant computational overhead associated with generating high-resolution images, [4] proposed a Latent Diffusion Model (LDM) where the diffusion model takes latent variables instead of images. Especially, the large-scale diffusion model (e.g., Stable Diffusion [4]) has leveraged LDM and learned from a large-scale dataset (LAION-Dataset [27]), enabling the creation of high-quality, high-resolution images conditioned on textual input.

## 2.2 Quantization

Quantization is a prominent neural network optimization technique that reduces storage requirements with low-precision representation and performance improvement when the corresponding low-precision acceleration is available. With $b-$bit precision, only $2^b$ quantization levels are accessible, which considerably limits the degree of freedom of the quantized model compared to floating-point representation. Therefore, the final quality significantly varies depending on the tuning of quantization hyperparameters, such as the quantization interval or zero offset. To preserve the quality of output in low-precision, it is crucial to update the quantization parameters as well as the model parameters, taking into account the specific features and requirements of the target task.

## 2.3 Quantization-aware Training and Post-training Quantization

Quantization algorithms can be broadly categorized into two types: Quantization-Aware Training (QAT) [19, 28, 29, 20, 30] and Post-Training Quantization (PTQ) [23, 24, 25]. QAT applies additional training after introducing quantization operators, allowing the update of network parameters toward minimizing the final loss value considering the effect of the quantization operators. On the other hand, PTQ does not apply end-to-end forward/backward propagation after quantization. Instead, it focuses on reducing block-wise reconstruction errors induced by quantization. QAT typically outperforms PTQ in low-precision scenarios, but it may not always be applicable due to limitations such as the deficiencies of datasets, training pipelines, and resource constraints. Due to its practical usefulness, PTQ has been actively researched recently. In the literature of diffusion models, [18] introduced a dedicated 8-bit post-training quantization method, demonstrating high-fidelity image generation performance.

On the other hand, while most QAT and PTQ algorithms focus on static quantization, recent research has highlighted the benefits of input-dependent dynamic quantization [31, 32, 33, 34]. Dynamic quantization enables the adjustment of quantization intervals based on the varying input-dependent distribution of activations, which has the potential to reduce quantization error. However, implementing these dynamic approaches often incur additional costs for extracting statistical information from activations, making it challenging to achieve performance improvements in practice. For instance, [35] pointed out that input-dependent activation functions can cause significant latency increases despite having relatively little computation.

While a number of previous works proposed different methods to accelerate sampling process of diffusion models, there have been limited works tried to exploit dynamic nature of diffusion models. In this paper, we propose a novel quantization scheme for diffusion models which minimizes activation quantization errors by enabling the generation of suitable quantization interval based on the time step information, all without incurring additional inference costs.

# 3 Temporal Dynamic Quantization

## 3.1 Quantization Methods

Before elaborating the proposed method, we define the quantization function used in our work. In this study, we focus on $b-$bit linear quantization, where the $2^b$ possible quantization levels are evenly spaced. Linear quantization involves two key hyperparameters: the quantization interval $s$ and the zero offset $z$. Given the full-precision data $x$, the quantized data $\hat{x}$ can be calculated as follows:

$$\bar{x} = clip(\lfloor x/s \rceil + z, n, p), \qquad \hat{x} = s \cdot (\bar{x} - z), \tag{5}$$

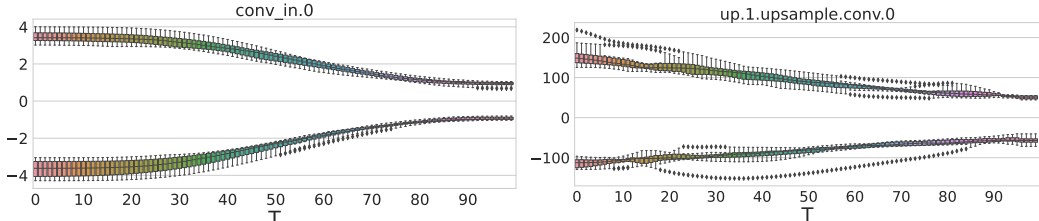

Figure 2: **Evolution of Activation Distribution of Diffusion Model over Time Steps**. Boxplot depicting the maximum and minimum values of activation for the DDIM model trained on the CIFAR-10 dataset across different time steps.

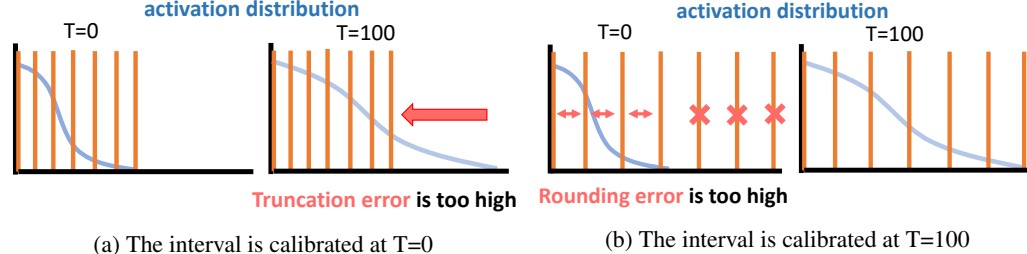

(a) The interval is calibrated at T=0        (b) The interval is calibrated at T=100

Figure 3: **Limitation of Static Quantization for Diffusion Models**. Assume that the activation distribution enlarges as time step progresses. (a) Large truncation error due to small quantization interval and (b) Large rounding error due to large quantization interval.

where $n(p)$ is the smallest(largest) quantization index, $clip(\cdot)$ is a clipping function and $\lfloor \cdot \rceil$ is a rounding function. For practical acceleration purposes, we utilize symmetric quantization for weights, where $z = 0$ and $p = -n = 2^{(b-1)} - 1$. On the other hand, for activation quantization, we assume asymmetric quantization with $z \neq 0$, $n = 0$, and $p = 2^b - 1$, which is a commonly adopted approach [36].

## 3.2   Challenges of Diffusion Model Quantization

The biggest challenge of quantizing diffusion models is to find optimal quantization parameters ($s$ and $z$) for activation that minimize the quantization error. As shown in Fig. 2, activation distribution of diffusion models have a unique property due to the iterative denoising process, whose distribution highly varies depending on the time step ($t$) regardless of the layer index.

Therefore, using static values for quantization parameters causes significant quantization error for different time steps, as shown in Fig. 3. Previous studies [37, 18] has also reported the dynamic property of activations in diffusion models and attempted to address it by sampling the calibration dataset across overall time frames. However, despite these efforts, these studies still relied on static parameters, resulting in sub-optimal convergence of minimizing quantization error. To tackle this problem fundamentally, it is crucial to enable the update of quantization parameters considering the dynamics in the input activation distribution.

## 3.3   Implementation of TDQ Module

To address the rapid changes in input activation, one easy way of think is input-dependent dynamic quantization. Previous studies have demonstrated that incorporating a quantization module that generates the quantization parameters based on input features such as minimum, maximum, and average values can lead to significant improvements in accuracy for specific applications [38, 32]. According to our observation, as will be presented in the Appendix, our own implementation of input-dependent dynamic quantization offers notable quality improvement. However, the process of gathering statistics on the given activation introduces complex implementation and notable overhead [35]. Despite the potential of quality improvement, this approach may not be an attractive solution in practice.

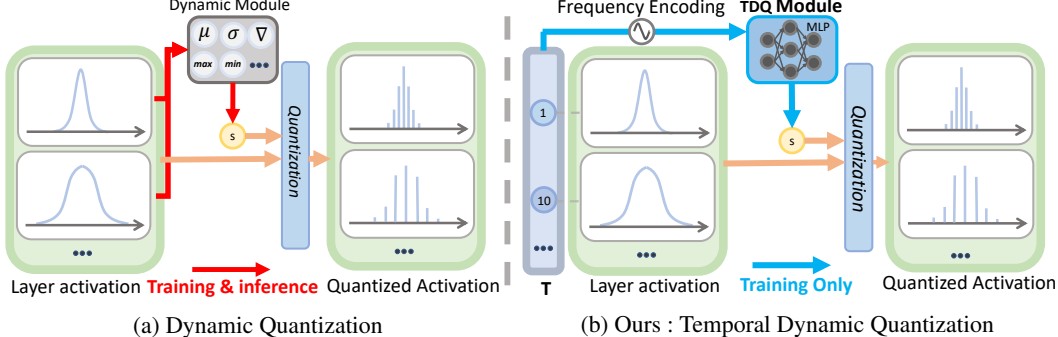

Figure 4: **Overview of TDQ module**. Comparison between (a) Input-dependent dynamic quantization, which requires activation statistics during both training and inference, and (b) the proposed TDQ module that enables cost-free inference based on pre-computed intervals.

Instead, we propose a novel dynamic quantization that utilizes temporal information rather than input activation. One important observation we made is that when we measured the Pearson correlation between time steps and per-tensor activation variation, 62.1% of the layers exhibited moderate temporal dependence ($|r| > 0.5$), and 38.8% exhibited strong temporal dependence ($|r| > 0.7$). This implies that many layers in diffusion models indeed possess strong temporal dependence. Although the activation distribution might change based on the input data, the overall trends remain similar within the same time frame (see Fig 2). Therefore, we can determine the optimal interval based on the differences in time steps, leading to more reliable and robust quantization results

Intuitively, we can employ a scalar quantization interval for each time step. However, this strategy falls short in capturing inter-temporal relationships and cannot be applied to scenarios where different time steps are used between training and inference. Instead, we propose using a learnable small network that predicts quantization interval by taking time step information. This strategy can evaluate activation alterations collectively across multiple timesteps, thereby making the learning process more stable and improving performance, as will be shown in Table 3.

Fig. 4 (b) presents the overview of our idea, called TDQ module. In the TDQ module, the dynamic interval $\tilde{s}$ is generated based on the provided time step $t$ as follows:

$$I = enc(t), \quad \tilde{s} = f(I), \tag{6}$$

where $enc(\cdot)$ represents the encoding function of the time step, which will be described in Section 3.4. Here, $I$ is the encoded feature, and $f(\cdot)$ signifies the generator module function.

In our approach, the TDQ module is attached to each quantization operator, and the independent dynamic quantization interval is generated based on the given time step. As illustrated in the Fig. 4, the generator is implemented by simply stacking multiple linear layers with a softplus function to constrain the data range to non-negative value. Please note that all components of the generator are differentiable. As a result, during the PTQ or QAT process, the interval can be updated toward minimizing quantization error using gradient descent.

For instance, when employing a well-known QAT algorithm such as LSQ [29], the static interval can be substituted by the output of the TDQ module. The quantization function from Eq. 5 is then modified as follows:

$$\bar{x} = clip(\lfloor x/\tilde{s} \rceil + z, n, p), \qquad \hat{x} = \tilde{s} \cdot (\bar{x} - z). \tag{7}$$

When we use a straight-through estimator [39], the gradient can propagate through the learnable parameters of the generator. These parameters are then updated via iterative gradient descent, with the goal of minimizing the final task loss. The same pipeline can be applied to PTQ algorithms that utilize local gradient of minimizing reconstruction error [23, 24]. Consequently, the TDQ module is easily applicable to both QAT and PTQ schemes.

However, unlike input-dependent dynamic quantization, the TDQ module enables cost-free inference. After the PTQ or QAT process, the time-dependent interval can be pre-computed offline, and during inference, we can utilize the pre-computed value. In addition, the pre-computed interval provides

a strong advantage by enabling the seamless integration of TDQ module on existing frameworks without modifications.

## 3.4 Engineering Details

We conducted a series of experiments to enhance the stability and effectiveness of the TDQ module, and figured out that several engineering improvements play a key role in achieving reliable results. This section provides a detailed description of these improvements and their impact on the performance of the TDQ module.

**Frequency Encoding of Time Step**    Feeding the time step directly to the generator results in inferior convergence quality. This is primarily due to the well-known low-frequency inductive bias of neural networks [40, 41]. If the time step is directly input to the generator, it tends to produce an interval that barely changes regardless of the time step. To mitigate this low-frequency bias, we use geometric Fourier encoding for time step [42, 43], as described in Eq. 6.

$$I = enc(t) = (sin(\frac{t}{t_{max}^{0/d}}), cos(\frac{t}{t_{max}^{0/d}}), sin(\frac{t}{t_{max}^{2/d}}), cos(\frac{t}{t_{max}^{2/d}}), ..., sin(\frac{t}{t_{max}^{d/d}}), cos(\frac{t}{t_{max}^{d/d}})), \quad (8)$$

where $t$ is the current time step, $d$ is the dimension of encoding vector, and $I$ is the encoded vector. This encoding approach allows the TDQ module to accommodate high-frequency dynamics of time step. In this paper, we set $t_{max}$ as 10000, empirically.

**Initialization of TDQ Module**    Proper initialization of quantization interval is crucial, as incorrect initialization can lead to instability in QAT or PTQ processes. Existing quantization techniques only need to initialize the static step value, but we need to initialize the TDQ module's output to the desired value. To achieve this, we utilize He initialization [44] for the weights and set the bias of the TDQ module (MLP)'s last linear layer to the desired value. Given that the input to the MLP (geometric Fourier encoding) can be treated as a random variable with a mean of zero, the output of the He-initialized MLP will also have a mean of zero. Thus, we can control the mean of the MLP output to a desired value via bias adjustment. After extracting 1000 samples for the entire time step, we initialized the quantization interval to minimize the overall error and then conducted an update to allow adaptation to each time step.

## 4    Experimental Setup

In order to demonstrate the superior performance of TDQ, we conducted tests using two different models: DDIM [2], a pixel space diffusion model, and LDM [4], a latent space diffusion model. For the DDIM experiments, we utilized the CIFAR-10 dataset [45] (32x32), while for the LDM experiments, we employed the LSUN Churches dataset [46] (256x256). This allowed us to showcase the effectiveness of the proposed method in both low and high-resolution image generation scenarios. We applied PTQ and QAT to both models. However, it is important to note that while the latent diffusion model consists of a VAE and a diffusion model, we focused on quantizing the diffusion model and did not perform quantization on the VAE component.

In the absence of prior QAT studies for the diffusion model, we experimented with well-known static quantization methods, i.e., PACT [19], LSQ [29], and NIPQ [30] as baselines. Our idea is integrated on top of LSQ, by replacing the static interval as an output of TDQ module. We also provide additional experiments of TDQ integration with other methods in the Appendix. Per-layer quantization was applied to activations and weights of all convolutional and linear layers, including the activation of the attention layer. The models were trained for 200K iterations on CIFAR-10 and LSUN-churches, with respective batch sizes of 128 and 32. The learning rate schedule was consistent with the full precision model.

In PTQ, we used PTQ4DM [18], a state-of-the-art study, as a baseline for extensive analysis. For a fair comparison, we mirrored the experimental settings of PTQ4DM but modified the activation quantization operator with a TDQ module for dynamic quantization interval generation. Like PTQ4DM, we used per-channel asymmetric quantization for weight and per-tensor asymmetric quantization for activation. The weight quantization range was determined by per-channel minimum/maximum values, and the activation quantization range was trained via gradient descent to minimize blockwise

Table 1: **QAT results of diffusion models**. N/A represents the failure of convergence.

| (CIFAR-10) | Methods | FID ↓ / IS ↑ | | | | |
|---|---|---|---|---|---|---|
| | | W8A8 | W4A8 | W8A4 | W4A4 | W3A3 |
| DDIM [2] | PACT [19] | 18.43 / 7.62 | 22.09 / 7.55 | 68.58 / 5.93 | 51.92 / 6.41 | N/A / N/A |
| | LSQ [29] | 3.87 / 9.62 | 4.53 / 9.38 | 6.2 / 9.56 | 7.3 / 9.45 | 7.63 / 9.38 |
| | NIPQ [30] | 3.91 / 9.49 | 13.08 / 9.7 | 6.36 / 9.59 | 30.73 / 8.94 | N/A / N/A |
| | Ours | **3.77** / 9.58 | **4.13** / 9.59 | **4.56** / 9.64 | **4.48** / 9.76 | **6.48** / 9.26 |

| (Churches) | Methods | FID ↓ | | | | |
|---|---|---|---|---|---|---|
| | | W8A8 | W4A8 | W8A4 | W4A4 | W3A3 |
| LDM [4] | PACT [19] | 9.20 | 9.94 | 8.59 | 10.35 | 12.95 |
| | LSQ [29] | N/A | 4.92 | 5.08 | 5.06 | 7.21 |
| | NIPQ [30] | 4.12 | 7.22 | **4.68** | 9.13 | N/A |
| | Ours | **3.87** | **4.04** | 4.86 | **4.64** | **6.57** |

Table 2: **PTQ results of diffusion models**. FP represents the output of the full-precision checkpoint.

| (CIFAR-10) | Methods | FID ↓ / IS ↑ | | | FP |
|---|---|---|---|---|---|
| | | W8A8 | W8A7 | W8A6 | |
| DDIM [2] | Min-Max | 8.73 / 9 | 34.61 / 8.28 | 332.15 / 1.47 | |
| | PTQ4DM [18] | 6.15 / 8.84 | 6.2 / 8.83 | 22.43 / 8.77 | 5.59 / 8.94 |
| | Ours | **5.99** / 8.85 | **5.8** / 8.82 | **5.71** / 8.84 | |

| (Churches) | Methods | FID ↓ | | | FP |
|---|---|---|---|---|---|
| | | W8A8 | W8A6 | W8A5 | |
| LDM [4] | Min-Max | 4.34 | 103.15 | 269.05 | |
| | PTQ4DM [18] | 3.97 | 4.26 | 7.06 | 4.04 |
| | Ours | **3.89** | **4.24** | **4.85** | |

reconstruction loss, as in methods like BRECQ [24]. Quantization was applied to all layers, but in the LDM PTQ experiment, the activation of the attention matrix was not quantized. Following PTQ4DM's approach, we used a calibration set of 5120 samples for PTQ, consisting of 256 images with 20 random time steps selected per image.

To measure the performance of the models, we used Fréchet Inception Distance (FID) [47] and Inception Score (IS) [48] for CIFAR-10, and FID for LSUN-churches. For evaluation, we generated 50,000 images using QAT with DDIM 200 step sampling, and 50,000 images using PTQ with DDIM 100 steps. In the case of QAT, we selected 5 checkpoints with the lowest validation loss and reported the scores from the best performing models.

All of experiments were conducted on the high performance servers having 4xA100 GPUs and 8xRTX3090 GPUs with PyTorch [49] 2.0 framework. The source code is available at `https://github.com/ECoLab-POSTECH/TDQ_NeurIPS2023`. Besides, we use the notation WxAy to represent x-bit weight & y-bit activation quantization for brevity.

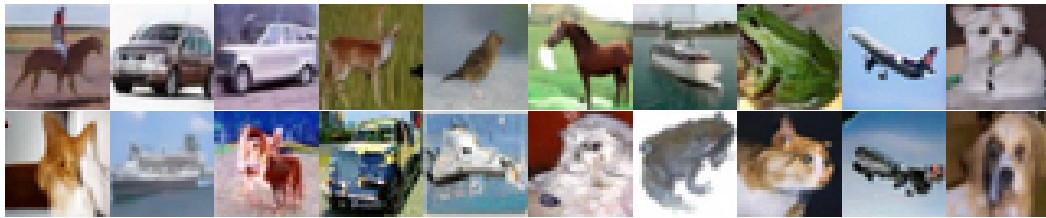

Figure 5: Visualization of W4A4 QAT results, DDIM on CIFAR-10 dataset, (up) Ours (Down) LSQ

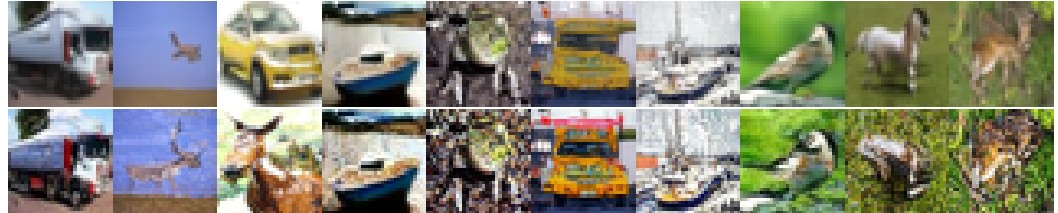

Figure 6: Visualization of W8A6 PTQ results, DDIM on CIFAR-10 dataset, (up) Ours (Down) PTQ4DM

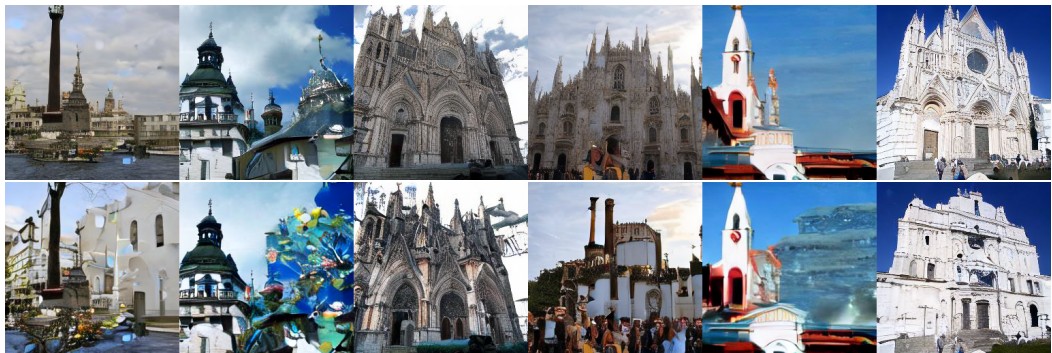

Figure 7: Visualization of W8A5 PTQ results, LDM on LSUN-churches dataset, (up) Ours (Down) PTQ4DM

## 5 Results

### 5.1 Quality Analysis after QAT and PTQ

Table 1 compares the TDQ module with existing static quantization methods. As the activation bit decreases, all static quantization methods show inferior quality, while our approach shows consistent output. TDQ module gives substantial quality improvement even in 8-bit, and the benefit becomes even large in 4-bit precision, showing negligible quality degradation to full-precision output. TDQ achieves these benefits by efficiently allocating the limited activation quantization levels.

Besides, NIPQ was introduced to address the instability of the straight-through estimator by applying pseudo quantization based on artificial noise. However, the noise of NIPQ is indistinguishable from the input noise, hindering the diffusion model's convergence. Additional efforts are required to exploit the benefit of PQN-based QAT for diffusion models.

Table 2 presents PTQ comparison of TDQ module against existing static PTQ schemes. The Min-Max method represents a naive linear quantization approach where the range is determined by the minimum and maximum values of the target tensor. The experiments demonstrate that while all baselines maintain a good level of FID when the activation bit is high, they experience significant performance degradation as the activation bit decreases. In contrast, TDQ exhibits only a slight level of FID degradation, indicating that the TDQ module is a robust methodology that performs well in both QAT and PTQ scenarios.

Fig. 5 to 7 visualize the generated images of quantized diffusion models. Our method consistently outperforms other quantization techniques in producing images with high-fidelity within the same bit-width configuration. Figs. 5 and 6 further illustrate this, as conventional QAT and PTQ produce blurred and unrecognizable images, whereas our method generates realistic images. The integration of temporal information in activation quantization is shown to be highly effective in maintaining the perceptual quality of the output.

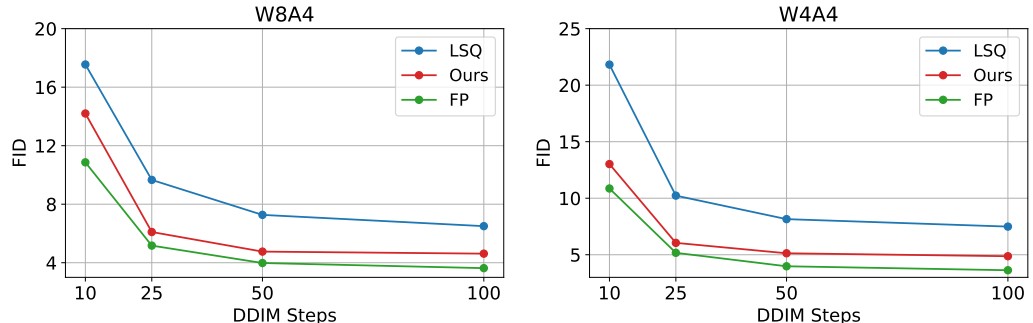

Figure 8: **Number of sampling step vs FID**. Comparison between the proposed method and conventional QAT when reducing the inference time step from 100 to 10.

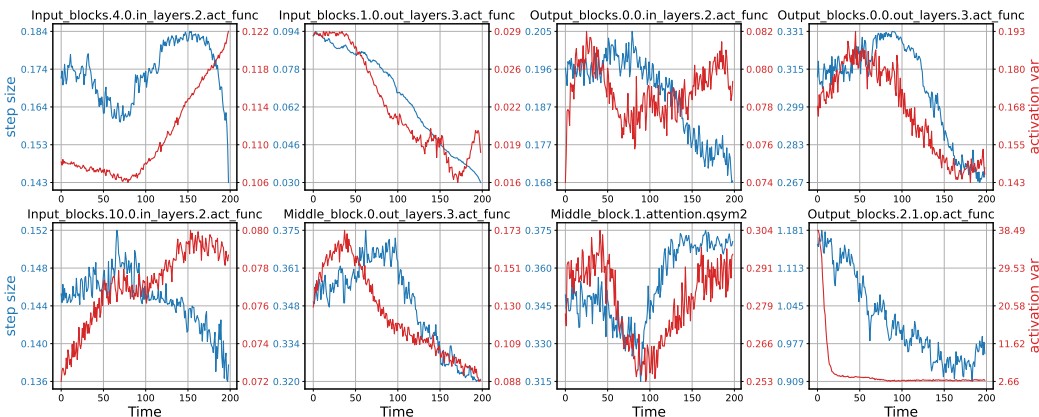

Figure 9: **Output dynamics of TDQ module**. Visualization of the generated interval versus the variation of input activation.

## 5.2 Generalization Performance of TDQ Module

This section presents the experimental results on the performance of the TDQ module for fast-forward inference. Training is carried out to fully encompass all time steps from 1 to 1000, however, inference can be executed with fewer time steps (between 50 to 100) for enhanced performance. This changes the distribution of the time steps during training/testing, thus, the TDQ module's generation requires good generalization abilities.

Fig. 8 displays the FID measurement for the DDIM model quantized by the LSQ algorithm for both W8A4 and W4A4 configurations. The time step gradually decreases from 100 to 10 during inference. As depicted, LSQ's performance significantly deteriorates as the number of time step decreases, whereas our method's performance declines similarly to the full-precision baseline. This experiment demonstrates that the TDQ module functions effectively even when the sampling time step varies.

## 5.3 Ablation Studies of TDQ module

To investigate the output dynamics of the TDQ module, we visualized the updates of the dynamic interval in relation to time steps (Fig. 9). The interval, trained using W4A4 LSQ on DDIM, demonstrates a tendency to change in alignment with activation variations. However, this pattern is not consistently observed across all layers. The inconsistencies could potentially indicate that the TDQ module is attempting to generate an interval that minimizes the final loss value, as LSQ adjusts the quantization interval accordingly.

To offer a more comprehensive analysis of the advantages of the TDQ module, we conducted a comparison with input-dependent dynamic quantization methods and performed an ablation study on the TDQ module. As depicted in Table 4, TDQ achieves superior performance compared to dynamic

Table 3: **Learning quantization interval directly for each time step.** The notation $S_N$ means that there are total of $N$ learnable quantization interval that cover 1000 / $N$ time steps uniformly.

| (QAT) | Bit-width | FID ↓ | | | | | |
|---|---|---|---|---|---|---|---|
| | | **Ours** | LSQ($S_1$) | $S_{10}$ | $S_{50}$ | $S_{100}$ | $S_{1000}$ |
| DDIM[2]-CIFAR10 | W4A4 | **4.48** | 7.30 | 4.75 | 5.17 | 4.88 | 5.53 |
| | W3A3 | **6.48** | 7.63 | 8.89 | 6.92 | 6.82 | 9.94 |
| LDM[4]-Churches | W4A4 | **4.64** | 5.06 | 5.17 | 5.07 | 4.78 | 5.10 |
| | W3A3 | 6.57 | 7.21 | 6.70 | **6.47** | 6.64 | 6.86 |

quantization, even without directly utilizing the layer's distribution information. We also investigated the effect of the number of layers within the TDQ module on performance. The results demonstrate that the output quality becomes substantially consistent when the number of layers exceeds 4. Based on these findings, we have empirically selected 4 layers of MLP, considering a balance of performance and complexity.

We also conducted experiments to compare the performance of TDQ with a configuration where the per-time quantization intervals are directly learned. As presented in Table 3, while the concept of directly learning the quantization interval for each time step and using a shared quantization interval across multiple time steps can enhance output quality compared to the baseline ($S_1$), TDQ continues to exhibit superior output quality and offers greater versatility. The success of TDQ lies in its ability to facilitate continuous and stable learning of the quantization interval while taking into account the evolution of the activation distribution over neighboring time steps.

Table 4: Ablation on TDQ module

| method | FID↓ |
|---|---|
| FP32 | 3.56 |
| Static Quantization(LSQ) | 7.3 |
| Dynamic Quantization (max) | 6.30 |
| TDQ (2 layer) | 5.18 |
| TDQ (4 layer) | **4.48** |
| TDQ (8 layer) | 4.88 |

## 6 Discussion

In this section, we discuss additional intuitions and limitations. While many layers exhibit strong temporal dependency, approximately 30% of the layers still do not show such correlations. These layers are primarily located in the middle block of the U-Net and are heavily influenced by their instance-wise semantic information. Moreover, we observed that the LDM model [4] has fewer temporal dependencies and thus demonstrates fewer performance improvements. However, it's crucial to emphasize that even in these cases, the TDQ module ensures convergence, resulting in consistent outputs across time steps. This alignment guarantees that the output quality remains comparable to that of existing PTQ/QAT algorithms, even in scenarios where temporal dependencies might be less.

## 7 Conclusion

In this paper, we explore the challenge of activation quantization in the diffusion model, specifically the dynamics of activation across time steps, and show that existing static quantization methods fall short of effectively addressing this issue. We introduce the TDQ module, seamlessly integrated into both QAT and PTQ, while enhancing the output quality significantly based on the dynamic quantization interval generation. Since our approach can be implemented without any overhead during inference, we expect our study to help improve performance while preserving quality in low-bit diffusion models for mobile and edge devices.

## Acknowledgements

This work was supported by IITP grant funded by the Korea government (MSIT, No.2019-0-01906, No.2021-0-00105, and No.2021-0-00310).

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
