# Supplementary Material for Temporal Dynamic Quantization for Diffusion Models

## 1 Introduction

In this Supplementary material, we present the results of the experiments mentioned in the paper, along with additional experiments. The following items are provided:

- A comparison between Dynamic Quantization and TDQ in Section 2.
- Ablation study on time step encoding in Section 3.
- Detailed TDQ Module architecture in Section 4.
- Comparison with multiple quantization interval directly on PTQ in Section 5.
- Integration of TDQ with various QAT schemes in Section 6.
- Various experiments about robustness of the TDQ Module in Section 7.
- Detailed experimental results on the Output dynamics of the TDQ module in Section 8.
- Detailed experimental results on the Evolution of Activation Distribution in Section 9.
- Various non-cherry-picked results of generated images in Section 10.

Table 1: Dynamic Quantization vs TDQ

| method | FID | IS |
|---|---|---|
| FP32 | 3.56 | 9.53 |
| Static Quantization(LSQ) | 7.3 | 9.45 |
| (a) Dynamic Quantization | 6.21 | 9.11 |
| (b) + Generator | 5.49 | 9.81 |
| (c) + Time information | 4.96 | 9.4 |
| TDQ | 4.48 | 9.76 |

Table 2: Time step encoding ablation

| Time embedding | FID | IS |
|---|---|---|
| No preprocess (T=1,2,...) | N/A | N/A |
| Normalize | 5.03 | 9.49 |
| random fourier | 4.76 | 9.53 |
| frequency encoding | 4.48 | 9.76 |

## 2 Dynamic Quantization VS TDQ

In Table 1, we compare the performance of Dynamic Quantization and TDQ on CIFAR-10 DDIM W4A4 QAT. In the implementation of Dynamic Quantization, three approaches were compared: (a) determining the quantization interval for each layer dynamically using the formula $s = x.abs().max()/(n\_lv)$, (b) passing layer statistics information (Min, Max, Mean, Var) to a trainable generator for interval generation, and (c) combining these statistics with time information for interval generation.

Notably, the results demonstrate that TDQ achieves superior performance compared to Dynamic Quantization, even without direct utilization of the layer's distribution information. Moreover, it is observed that when time information and distribution information are used together, the performance is better when utilizing time information alone. We believe that this is because directly using the distribution information of each layer to determine the quantization interval can result in suboptimal performance due to the sensitivity of the generated intervals to rapid changes in the layer's distribution, such as outlier data.

37th Conference on Neural Information Processing Systems (NeurIPS 2023).

# 3 Ablation Study of Time Step Encoding

In this section, an ablation study was conducted on the time step encoding, which serves as the input for the TDQ module. Table 2 presents the ablation study on the methods for encoding time step information into an TDQ module. The experiment is conducted on CIFAR-10 DDIM W4A4 QAT. As observed from the table, when no preprocessing is applied, the training becomes infeasible due to significant differences in values. Training stabilizes when normalization is applied; however, it is noted that encoding techniques such as frequency encoding or random Fourier encoding result in improved performance by incorporating high-frequency inductive biases. Although Random Fourier encoding exhibits slightly superior performance, it introduces additional training costs. Hence, a simpler and deterministic frequency encoding approach was employed.

# 4 TDQ Module Architecture

| Layer | Dim |
|---|---|
| Linear | 128 x 64 |
| ReLU | - |
| Linear | 64 x 64 |
| ReLU | - |
| Linear | 64 x 64 |
| ReLU | - |
| Dropout | (0.2) |
| Linear | 64 x 1 |
| Softplus | - |

Table 3: Overview of TDQ Module architecture

As shown in Table 3, TDQ module consists of a carefully designed 4-layer MLP with ReLU activation. The module takes time embedding as input and has 4 hidden layer with 64 dimensions. One dropout layer is included before the last linear layer to prevent overfitting. In our experiments, we used a dropout ratio of 0.2. For the last layer, we constrain the output to produce positive values using Softplus function, since the quantization interval is always positive.

# 5 Leaning Multiple Quantization Interval Directly on PTQ

Table 4: Performance comparison with learning multiple quantization interval on PTQ

| FID↓ | TDQ | $TDQ_{thin}$ | $S_1$[1] | $S_2$ | $S_4$ | $S_{10}$ | $S_{20}$ |
|---|---|---|---|---|---|---|---|
| Churches-LDM W4A8 | 44.88 | **28.74** | 76.36 | N/A | 33.19 | 37.63 | 29.87 |
| Churches-LDM W4A6 | 120.53 | 55.27 | 158.07 | 193.11 | 185.19 | 72.37 | **47.39** |
| ImageNet-LDM W4A6 | 41.23 | **16.96** | 47.26 | N/A | 85.11 | 21.71 | 17.38 |

In this section, we compare with strategy that learn quantization interval directly and Q-Diffusion($S_1$) [1] in PTQ setting. We borrow codebase from PTQD [2] for this experiment. For brevity, the notation $S_N$ signifies that there are a total of $N$ learnable quantization interval parameters that cover 1000 / $N$

time steps uniformly. In Table 4, while the standard TDQ module yielded a notable performance boost over $S_1$, it lagged behind configurations such as $S_{20}$. Our investigations revealed that the default TDQ module was prone to overfitting due to two primary reasons: 1) Training the MLP with a limited calibration dataset (typically 256 samples) proves challenging. 2) The relatively brief training iteration, typically 5000 in BRCEQ [3], finds it hard to filter out the high-frequency component present in frequency-encoded input features. To mitigate these constraints, we introduced a streamlined version of TDQ, referred to as $TDQ_{thin}$. This refined module uses a 3-layer MLP with a mere 16 hidden dimensions and omits the frequency encoding for time steps. Notably, this adaptation showcased results that either surpassed or matched the best performances observed in the $S_N$ experiments.

## 6  Integration of TDQ with various QAT schemes

In this section, we show the potential of the integration of TDQ with other existing QAT methods other than LSQ. Experiments were conducted on DDIM CIFAR-10 with 200 steps of sampling. As shown in Table 5, NIPQ [4], DuQ [5], and QIL [6] show significant performance improvements when combined with TDQ. On the other hand, PACT shows a performance degradation when used with TDQ. This is due to the fact that PACT uses L2 regularization to learn quantization intervals, rather than reducing quantization errors.

Table 5: TDQ with other QAT methods

| FID↓ | PACT [7] | NIPQ [4] | DuQ [5] | QIL [6] |
|---|---|---|---|---|
| baseline | **51.92** | 30.73 | 7.88 | 9.85 |
| +TDQ | 74.56 | **24.27** | **4.46** | **7.51** |

## 7  Robustness of the TDQ Module

In this section, we evaluate the robustness of TDQ module for training. Firstly, we perform three trials with distinct random seeds for both DDIM CIFAR-10 QAT and LDM churches QAT. As shown in Table 6, TDQ consistently outperforms conventional QAT methods, indicating its enhanced stability. Furthermore, an intriguing observation across various trials is the convergence of quantization intervals. In Fig. 1, these intervals tended to approach nearly identical values at each time step. This consistency further underlines TDQ's robustness. In Table 7, TDQ exhibits comparable accuracy across different random initialization methods, except for Gaussian initialization. These results indicate that TDQ module is relatively insensitive to other design choices.

Table 6: Experiment on three distinct seeds

| FID↓ | CIFAR-10 | | Churches | |
|---|---|---|---|---|
| | DDIM W4A4 | DDIM W3A3 | LDM W4A4 | LDM W3A3 |
| LSQ [8] | 7.13 ± 0.79 | 8.34 ± 0.71 | 5.30 ± 0.04 | 7.14 ± 0.07 |
| TDQ | **4.71 ± 0.16** | **6.67 ± 0.19** | **4.92 ± 0.28** | **6.83 ± 0.11** |

Table 7: TDQ Weight Initialzation

| FID↓ | He (Default) | Gaussian | Zero | Xavier |
|---|---|---|---|---|
| CIFAR-10 W4A4 | **4.48** | Fail | 4.98 | 4.90 |

Figure 1: Learned interval of TDQ with Different Random seeds

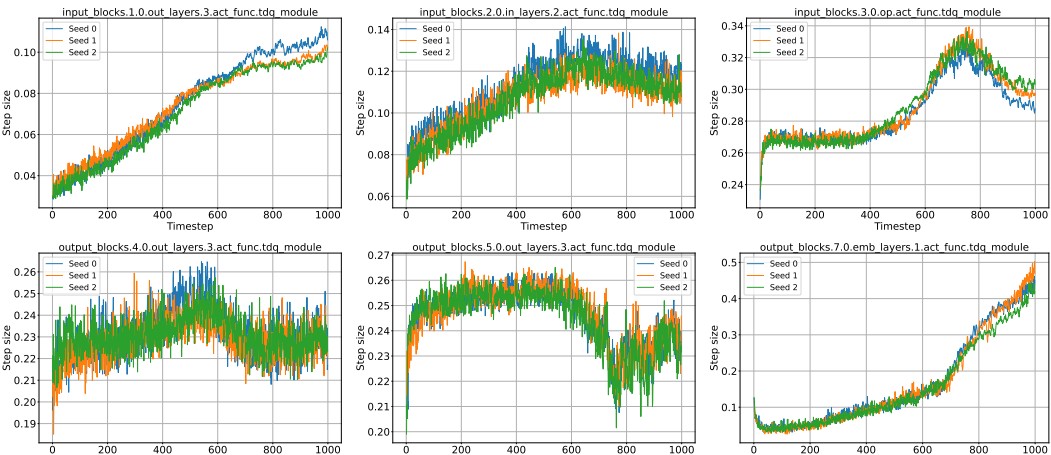

# 8 Output dynamics of TDQ module

In this section, we provide dynamics of TDQ module in more different layers and models. In Fig. 2, 3, the blue curve represents the quantization interval for each time step, while the red curve illustrates the fluctuations in the magnitudes of the activations over each time step. As expected, the ideal behavior is for the quantization interval to expand as the variance amplifies. This expansion ensures that quantization errors are minimized. For many layers, this desired behavior is evident. For instance, Input_blocks.1.0.emb_layers.1.act_func in Fig. 2 clearly demonstrates this pattern. However, certain layers, such as Middle_block.0.out_layers.3.act_func don't exhibit a pronounced effect. This discrepancy arises primarily due to the combination of significant variations in data and the layer's low temporal dependency. In scenarios like these, the output from TDQ tends to stabilize, converging to a specific value. This behavior mirrors that of LSQ, where the quantization interval becomes essentially constant.

# 9 Evolution of Activation Distribution

In Fig. 4, we demonstrate the different layer's temporal evolutions of the activation distribution trained on the CIFAR-10 DDIM model and text-conditioned Stable Diffusion.

In the DDIM model, we can observe that most layers possess temporal dependencies, especially showing high temporal dependency in input and output blocks. However, layers located in the middle block of the U-Net are heavily influenced by their instance-wise semantic information and thus have less temporal dependency and higher variance.

In Stable Diffusion, we extracted 12 images for three distinct text-prompts from Stable-Diffusion and depicted their activation distributions in Fig. 5. As revealed by the figure, input distributions display minor variations depending on text conditions. However, the feature map's overall trends seem to be more reliant on time than the specific text condition. Thus, we have found that our method can be applied to text-conditioned scenarios such as Stable-Diffusion without hindrance. The extension of the TDQ Module to cater to text conditioning presents an intriguing aspect and points to a promising avenue for future research.

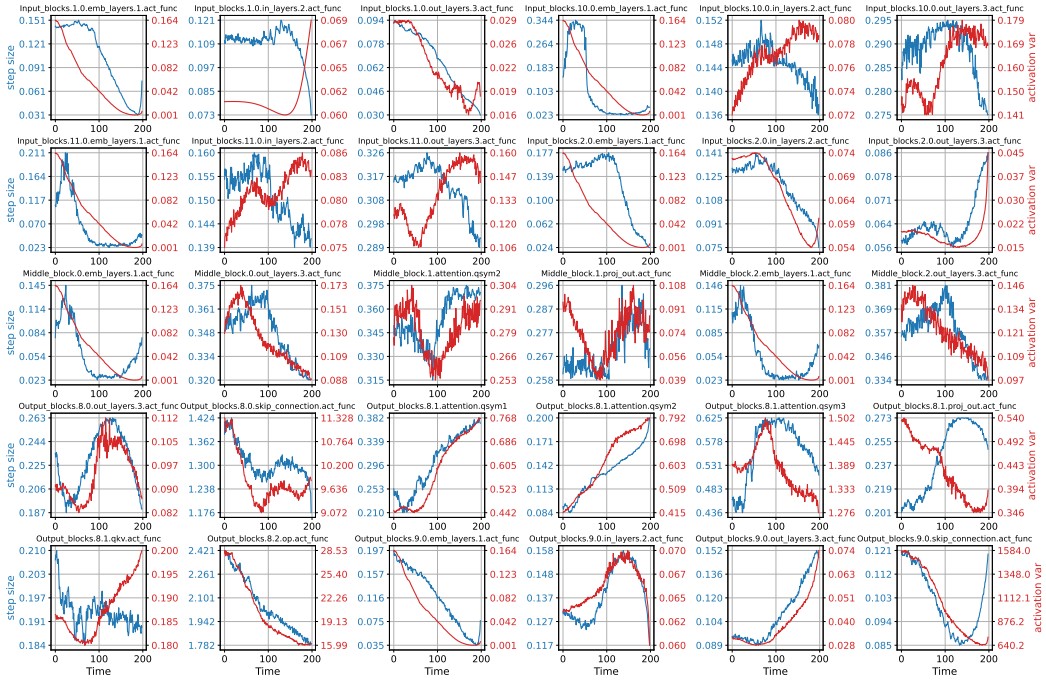

Figure 2: DDIM CIFAR-10 TDQ output dyanmics

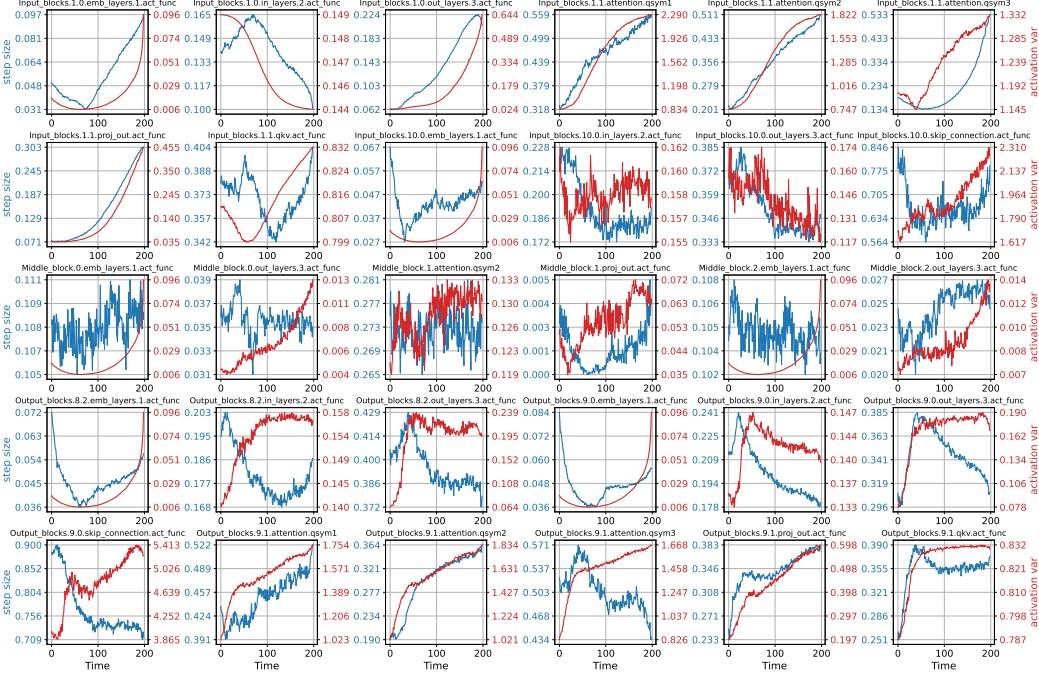

Figure 3: LDM LSUN-churches TDQ output dyanmics

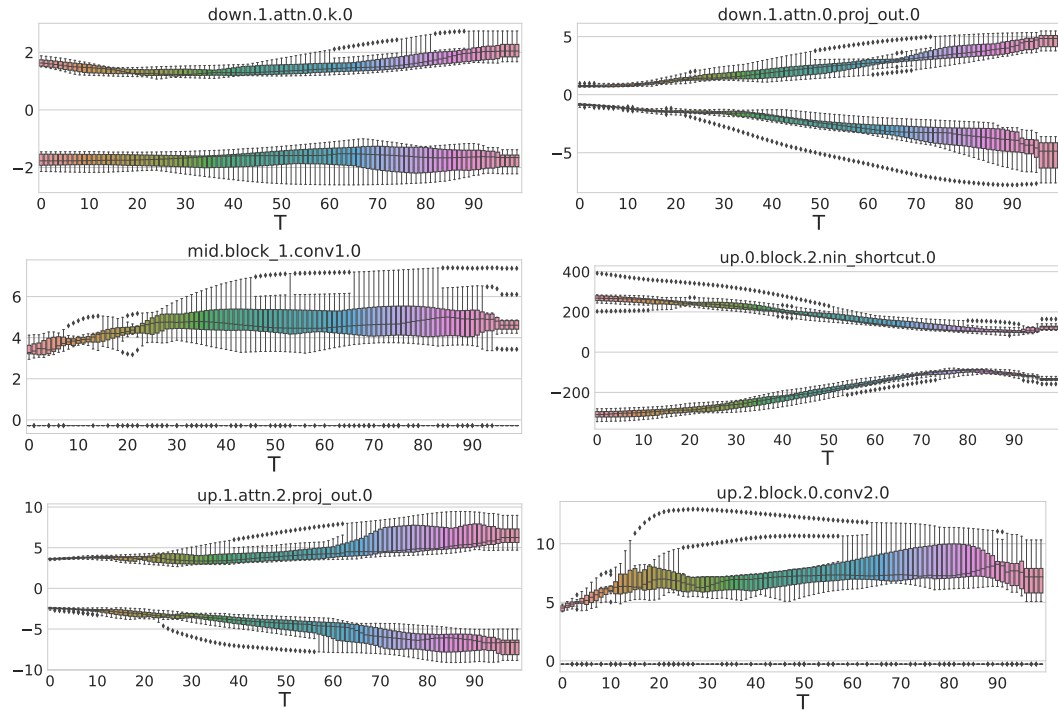

Figure 4: Temporal Evolution of Activation Distribution (DDIM CIFAR-10)

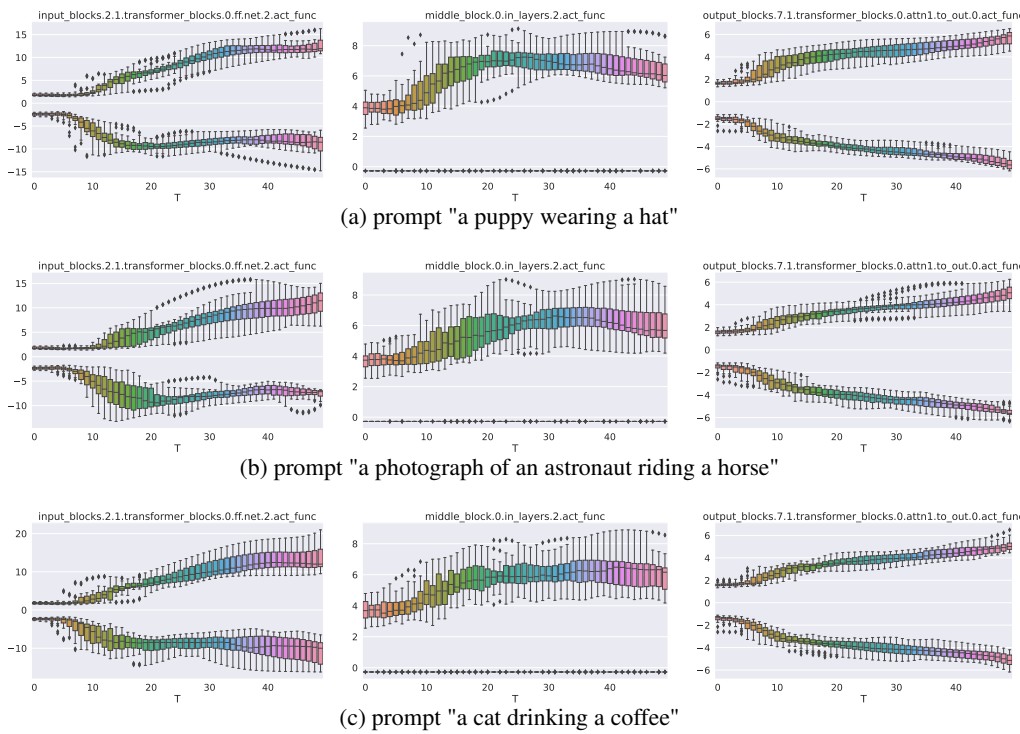

(a) prompt "a puppy wearing a hat"

(b) prompt "a photograph of an astronaut riding a horse"

(c) prompt "a cat drinking a coffee"

Figure 5: Text-Conditioned Temporal Evolution of Activation Distribution (Stable Diffusion)

## 10 Random Sampling Results

In this section, we provide our non-cherry-picked generated images using TDQ, under 4 different bitwidth setting(W8A8, W8A4, W4A8, W4A4). Results are shown in the Fig. 6, 7, 8, 9

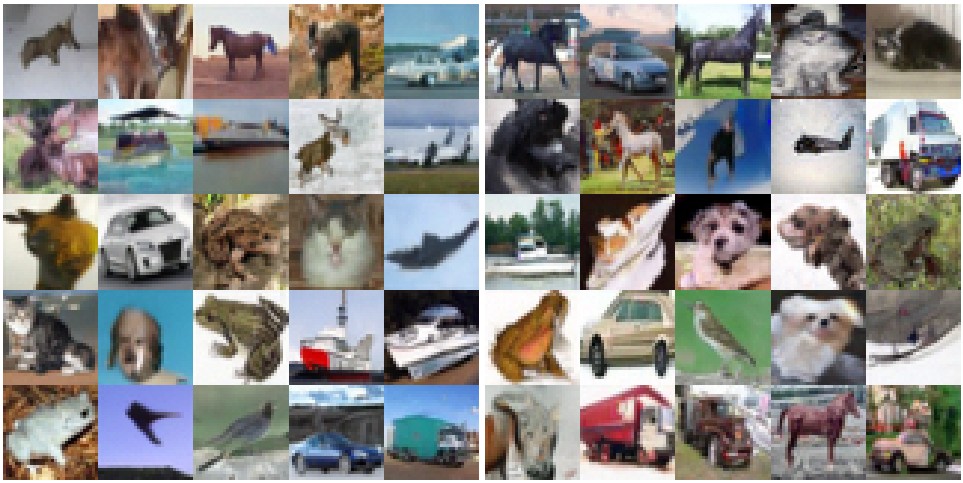

Figure 6: QAT CIFAR-10 Random Sampled Images W8A8 (Left) W8A4 (Right)

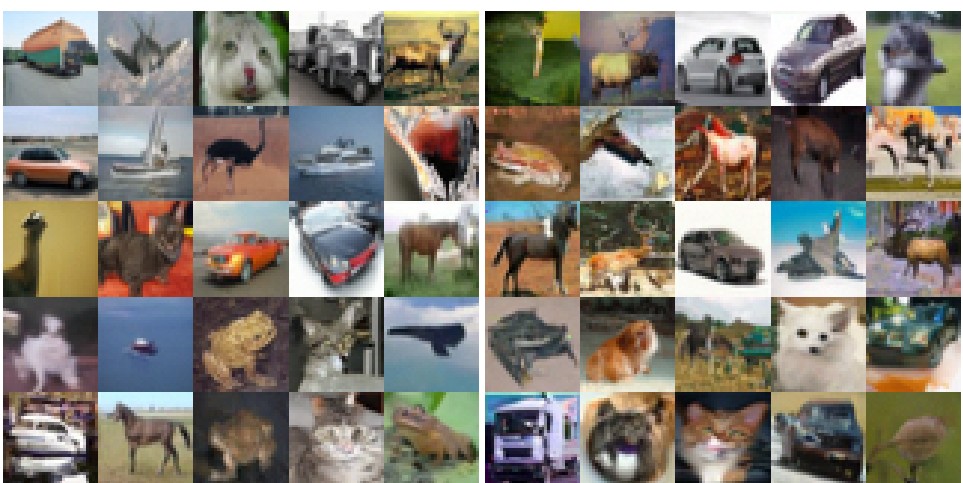

Figure 7: QAT CIFAR-10 Random Sampled Images W4A8 (Left), W4A4 (Right)

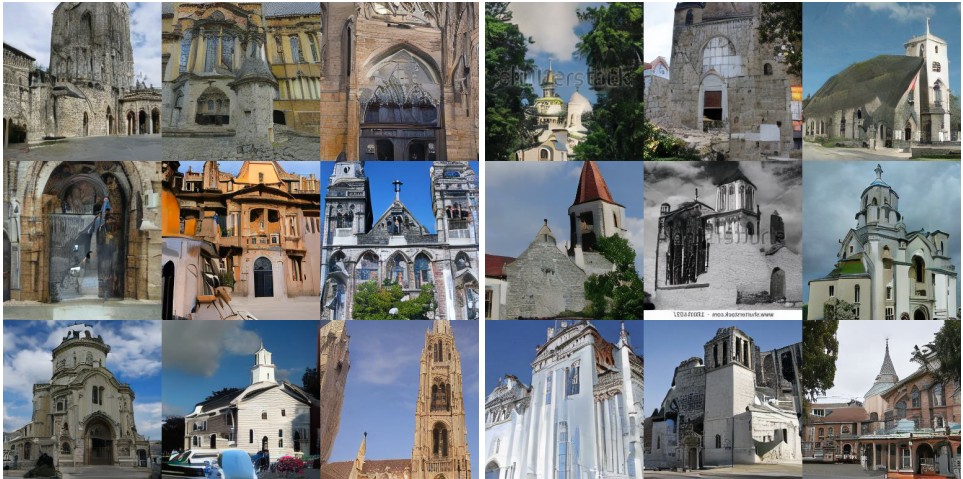

Figure 8: QAT LSUN Random Sampled Images W8A8 (Left) W8A4 (Right)

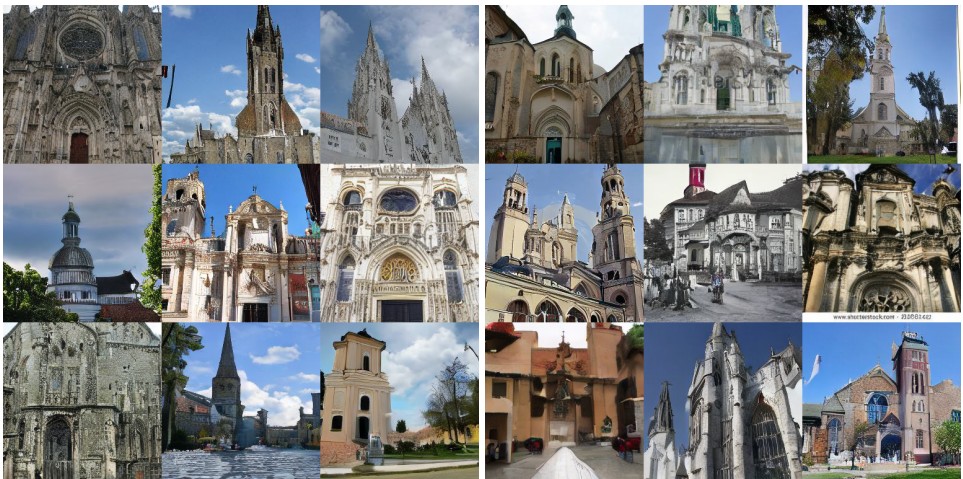

Figure 9: QAT LSUN Random Sampled Images W4A8 (Left), W4A4 (Right)