# OpenReview forum: "Temporal Dynamic Quantization for Diffusion Models"
_NeurIPS.cc/2023/Conference — NeurIPS 2023 poster_

### Official Review · Reviewer_ijsE · 2023-07-01

**Soundness:** 3 good
**Presentation:** 3 good
**Contribution:** 3 good
**Rating:** 5
**Confidence:** 4

**Summary:**

This paper presents a novel quantization method for diffusion models that dynamically adjusts the quantization interval. To this end, the authors present a unique Temporal Dynamic Quantization (TDQ) module, which utilizes the time step as an input to determine the quantization interval, which significantly improves the generation quality. Extensive experiments on CIFAR-10 and LSUN Churches demonstrate the superior performance of the proposed method.

**Strengths:**

The main idea is well-written and easy to follow. The proposed method, though simple in design, proves to be remarkably effective, notably diminishing the decline in performance. Its simplicity does not compromise its efficacy, making it a compelling solution to the problem at hand.

**Weaknesses:**

The primary weaknesses of the paper include a lack of a vital comparison with Q-diffusion, a key baseline in diffusion models, and a need for more comprehensive ablation studies, such as investigating the impact of the number of layers in the generator module on the model's overall performance.

**Questions:**

1.	In Section 3.1, the authors state that uses symmetric quantization for weights and set $p=-n=2^{b-1}-1$. However, compared with LSQ that set $n=-2^{b-1}$, it appears that the proposed method might be under-utilizing one quantization level. This could potentially lead to a slight inefficiency in the model’s performance.

2.	In Section 3.3, the authors state that previous studies have demonstrated that incorporating a quantization module … can lead to significant improvements in accuracy for specific applications. However, there appears to be a lack of citation or reference supporting this assertion.

3.	Instead of using the proposed TDQ module, it is possible to directly learn distinct quantization intervals corresponding to different time steps?

4.	The paper lacks a comparison in its experimental section. Specifically, the study has not considered Q-diffusion (as detailed in reference [36]) as a baseline. Since Q-diffusion also applies post-training quantization to diffusion models, it's critical for the authors to include a comparative analysis of their proposed method against Q-diffusion. This would help to further validate the efficacy of their approach and enrich the overall discussion.

5.	The paper could benefit from additional ablation studies, which would provide a more comprehensive understanding of the proposed method. One specific point of interest mentioned is the influence of the number of layers in the generator module on the overall performance. By investigating this aspect, it would add depth to the analysis and potential insights into the efficiency of the proposed model.


**Limitations:**

The authors do not provide any discussions on limitations.

---

> ### Author Rebuttal · Authors · 2023-08-09
>
> Thank you for your positive comment and insightful feedback.
>
> ---
>
> ### Weaknesses
>
> **W1: Comparsion with Q-Diffusion**
>
> We have thoroughly prepared a response to address your concern. Kindly refer to the global response section 2 where we detail the comparison with Q-Diffusion.
>
> ### Questions:
>
> **Q1: Rationale behind using \( $p = -n = 2^{b-1} - 1$ \)**
>
> Thank you for the keen observation. Indeed, in a pure mathematical context, the smallest integer level could be represented as $-2^{b-1}$ . However, it's essential to highlight that many prominent acceleration libraries, such as XNNPack in TensorFlow Lite, deliberately exclude the smallest integer value to maintain the symmetry of the data distribution. Our approach adheres to this widely accepted convention seen in quantization acceleration frameworks. Besides, please note that we apply the same restriction for LSQ algorithm where $n $ is set to $-2^{b-1} + 1 $ for fair comparison.
>
> **Q2. Citing dynamic quantization schemes for quality enhancement**
>
> Thank you for drawing attention to this aspect. As highlighted in reference [1], utilizing a dynamic computation of the token-wise Min-Max range has led to significant improvements in accuracy. Similarly, reference [2] underscores the enhancement in output quality, achieved by factoring in the Minimum, Variance, and Skewness of individual image instances during the quantization process. We will duly incorporate these references into our paper for a more comprehensive perspective.
>
> **Q3. Learning quantization interval directly for each time step**
>
> We have thoroughly prepared a response to address your concern. Kindly refer to the global response section where we detail the comparison with the direct interval learning for each time step
>
> **Q4. Comparison with Q-Diffusion**
>
> We have thoroughly prepared a response to address your concern. Kindly refer to the global response section where we detail the comparison with Q-Diffusion.
>
> ****Q5. Ablation study for the number of layers in the TDQ module****
>
> We appreciate your interest in understanding the effect of the number of layers within the TDQ module on the performance. In response to this, we conducted an ablation study to examine the impact of changing the number of layers within the TDQ module.
>
> The results, as detailed in the table below, illustrate that the output quality becomes substantially consistent when the number of layers exceeds 4. Based on these experimental findings, we have empirically selected 4 layers of MLP for the TDQ module. This decision was driven by a balanced consideration of performance and complexity.
>
> We recognize the importance of this analysis in enhancing the understanding of the TDQ module's design, and we will include the complete details and additional ablations in the camera-ready version of our paper.
>
> - **TDQ Number of layer ablation**
>   | (FID↓) | 2 Layer  | 4 Layer (Default) | 8 Layer |
>   | --- | --- | --- | --- |
>   | CIFAR10 W4A4 | 5.18 | **4.80** | 4.88 |
>
> ### Limitations
>
> **L1: Discussion on the limitations of our study**
>
> To elucidate the findings and potential limitations of our research, we have outlined key points below:
>
> 1. **Temporal dependency across layers**: The majority of layers in our study manifest the anticipated behavior where the activation distribution exhibits a temporal evolution. A salient representation of this is the layer **`Input_blocks.1.0.emb_layers.1.act_func`**, as highlighted in Appendix Figure 6, which lucidly captures the dynamics of the activation distribution over time. Conversely, specific intermediate layers, most notably **`mid.attn_1.proj_out.0`** (referenced in Figure 7 of the supplementary material), deviate from this behavior. The absence of marked temporal dependency in such layers stems chiefly from pronounced data variations and the intrinsically weak temporal relationship of the layer. Under these circumstances, the output from the TDQ module tends to stabilize, yielding a largely uniform value. This behavior parallels the tendencies exhibited by LSQ, where the quantization interval predominantly retains a constant magnitude.
> 2. **Issues of overfitting with the TDQ module during PTQ**: Subsequent to the review process, we observed that the default TDQ module is prone to overfitting during Post-training Quantization (PTQ), leading to a diminished output quality. This observation is further elaborated in the section titled "Learning quantization interval directly for each time step" in the global response. Our analysis suggests that employing a thin TDQ module, achieved by trimming the layer count to 3 and curtailing the hidden dimension to 16, can substantially enhance PTQ performance. We aim to incorporate this important observation in the forthcoming camera-ready iteration of our paper.
>
> ### References
>
> [1] Yao, Zhewei, et al. "Zeroquant: Efficient and affordable post-training quantization for large-scale transformers." Advances in Neural Information Processing Systems 35 (2022): 27168-27183.
>
> [2] Lee, Changhun, et al. "INSTA-BNN: Binary neural network with instance-aware threshold." *arXiv preprint arXiv:2204.07439*(2022).

---

> > ### Comment · Reviewer_ijsE · 2023-08-15
> > **Response to Authors**
> >
> > Dear Authors,
> >
> > Thank you for the comprehensive revisions and the detailed response to the previous comments. I commend your additional efforts, particularly the inclusion of new experiments and tables, which have undoubtedly enhanced the overall quality of the paper.
> >
> > However, upon closer examination, I still have some reservations concerning the experimental results. Specifically, in Tables 1 and 2, I noticed that the performance of QAT (w8a8) on Churches appears to be inferior to that of PTQ, despite utilizing a higher number of DDIM steps (200 compared to 100). Could you please shed some light on this observation?
> >
> > Looking forward to your clarification.
> >
> > Best regards,
> >
> > Reviewer ijsE

---

> > > ### Author Response · Authors · 2023-08-17
> > > **Response to Reviewer**
> > >
> > > We deeply appreciate your thoughtful feedback.  Each comment we receive plays a crucial role in refining the quality of our work.
> > >
> > > In response to your feedback, we concur that the results presented in Tables 1 and 2 for the QAT require revision.Subsequent to our paper's submission, we identified an error in our source code. Specifically, when implementing QAT on the pretrained model, it is essential to construct a new PyTorch model embedded with the quantization operator and then import the pretrained parameters from the traditional full-precision model. Regrettably, during this transition, one key parameter - the pretrained scale-factor of the LDM Autoencoder - was not loaded accurately. This oversight meant that during the QAT operation, certain parameters were initialized from random values rather than from the pretrained checkpoint, leading to a decrease in accuracy.
> > >
> > > Below, we present the corrected results of the QAT church experiment, which now incorporates the accurate scale-factor. In our ongoing commitment to comprehensive and fair evaluation, especially when juxtaposed with PTQ, we have utilized a 100-DDIM step for these updated results.
> > > |  | w8a8 | w4a8 | w8a4 | w4a4 | w3a3 |
> > > | --- | --- | --- | --- | --- | --- |
> > > | PACT | 9.20 | 9.94 | 8.59 | 10.35 | 12.95 |
> > > | LSQ | - | 4.92 | 5.08 | 5.06 | 7.21 |
> > > | TDQ | **3.87** | **4.04** | **4.86** | **4.64** | **6.57** |
> > >
> > >
> > > From the provided table, it's evident that QAT consistently outperforms PTQ, despite every attention layer being quantized. For those keen on replicating our results, please adjust the scale_factor to `0.2458` on `line 18` within the following file paths: `configs/lsun_churches-ldm-kl-8.yaml` and `configs/LSUN-churches_sampling.yaml`.
> > >
> > > It's also worth emphasizing that the benefits of our proposed method remain evident even in the revised results. While the entire suite of QAT experiments was impacted by the aforementioned error, a substantial proportion of the network parameters were correctly initialized. As a result, any degradation experienced was relatively minor and consistent across the board, but the overarching trend remains discernible.
> > >
> > > We deeply regret and apologize for any misunderstanding or confusion stemming from our initial erroneous experimental data. Rest assured, the corrected findings will be incorporated into the final draft of our paper.

---

> > > > ### Comment · Reviewer_ijsE · 2023-08-20
> > > > **Response to Authors**
> > > >
> > > > Dear Authors,
> > > >
> > > > I appreciate the thoroughness of your rebuttal and the additional clarifications provided. The supplementary results have addressed my previous concerns.
> > > >
> > > > Warm regards,
> > > >
> > > > Reviewer ijsE

---

### Official Review · Reviewer_xsTz · 2023-07-03

**Soundness:** 3 good
**Presentation:** 3 good
**Contribution:** 3 good
**Rating:** 6
**Confidence:** 3

**Summary:**

While diffusion models provide outstanding quality and diversity of generated data, they demand high storage and computation, limiting its applications on mobile devices. Further, the paper points out that current existing works on quantization are not suitable for diffusion models as diffusion models have unique temporal properties due to iterative sampling process. As such, the paper introduuces a new quantization technique for diffusion models, where quantization interval is dynamically adjusted. As a result, the proposed framework prvoides strong performance.

**Strengths:**

- The proposed method can be easily integrated with the existing QAT and PTQ algorithms.
- Clear visualizations which help the understanding of the paper and motivations.
- Simple yet effective quantization method without additional inference costs.

**Weaknesses:**

- Don't input-dependent quantization methods [31,32,33,34] provide computational efficiency benefits even after considering extra overhead? Is there a proof for supporting the following claim "dynamic approaches often incurs additional costs for extracting statistical information from activations, making it challenging to achieve performance improvements in practice."? It would be good to refer to the results or references for supporting this claim or tone down the claim a bit.
.
- Missing experimental comparisons against [36], which is another work on quantization of diffusion model.

- The paper claims that the activation distribution is similar across inputs with Figure 2. But, Figure 2 shows the activations of an earlier convolutional layer, which is known to have low-level features that are common across diverse layers. It would be good to show that similar trends can be observed across different layers. Indeed, less clear and less time dependent trends can be observed in deeper layers (mid.attn_1.proj_out.0, down.1.downsample.conv.0, down.1.attn.0.k.0, mid.block_1.conv1.0) in Figure 7 of the supplementary material. I suspect that deeper layers will exhibit even more complex trends that may not look like time-dependent. More detailed discussions are needed.

- The paper claims that the proposed method can be seamlessly integrated into existing quantization algorithms, but the paper only shows the benefit of doing that with LSQ in the case of QAT.

- Non-exhaustive list of typos (a thorough proof-reading is needed):
   - can decipts in Line 148
   - there have been limited works tried to in Line 109

**Questions:**

Please refer to the weakness section

**Limitations:**

No limitations mentioned in the paper. The potential limitation of the work is not taking into account for activation distribution variation that exists among different inputs.

---

> ### Author Rebuttal · Authors · 2023-08-09
>
>
> Thank you for your positive comment and constructive feedbacks
>
> ---
>
> ### Weaknesses
>
> **W1: Extra overhead of input-dependent quantization method**
>
> We appreciate the feedback and understand the concerns raised. We would like to direct the reviewer's attention to the recently published paper, "MobileOne."[1]. This paper highlights that input-dependent activation functions, such as Dynamic Shift-Max and Dynamic ReLU, experience latency that is more than 50 times slower compared to conventional activation functions like ReLU. Despite the relatively small number of operations these functions involve, the computation of input-dependent statistics greatly intensifies memory access, resulting in a significant increase in latency. Given that input-dependent quantization operators also leverage similar input-dependent statistics, latency becomes a predominant bottleneck. Nevertheless, we acknowledge the need for clarity and will revise our statement to be more measured and nuanced.
>
> **W2: Comparison with Q-Diffusion**
>
> We have thoroughly prepared a response to address your concern. Kindly refer to the global response section 2 where we detail the comparison with Q-Diffusion.
>
> **W3: Time dependency in deeper Layers**
>
> We acknowledge the feedback and appreciate the depth of your observation. For many layers, the expected behavior, where the activation distribution changes over time, is indeed prominent. As an illustrative example, the layer **`Input_blocks.1.0.emb_layers.1.act_func`**, depicted in Appendix Figure 6, vividly demonstrates the time-evolving nature of the activation distribution.
>
> However, as rightly pointed out by you, certain intermediate layers, notably **`mid.attn_1.proj_out.0`** (as referenced in Figure 7 of the supplementary material), do not exhibit a similar trend. This lack of temporal dependency in such layers can be attributed mainly to a combination of significant data variation and the inherent low temporal dependency of the layer itself. In situations of this kind, the TDQ output tends to converge, effectively stabilizing to produce a consistent value. This phenomenon is reminiscent of the behavior observed with LSQ, wherein the quantization interval largely remains static.
>
> **W4: Integration of TDQ with various QAT schemes**
>
> To investigate the general applicability of the TDQ approach across different Quantization-Aware Training (QAT) methods, we conducted further experiments with four QAT schemes: PACT, NIPQ, DuQ, and QIL. The training was conducted on CIFAR-10 DDIM W4A4 QAT. The results, as summarized in the table below, reveal that NIPQ, DuQ, and QIL demonstrate marked improvements when integrated with the TDQ scheme, underscoring the adaptability of TDQ with various gradient-based QAT methodologies.
>
> - **Integration with other quantization scheme**
>
>   | (FID↓) | PACT | NIPQ | DuQ | QIL |
>   | --- | --- | --- | --- | --- |
>   | baseline | **51.92** | 30.73 | 7.88 | 9.85 |
>   | +TDQ | 74.56 | **24.27** | **4.46** | **7.51** |
>
> However, an exception was observed with PACT, where the integration with TDQ led to a degradation in quality. This anomaly can be attributed to the specific nature of PACT, where the quantization interval is updated based on artificial L2 regularization, rather than through the minimization of quantization error. This divergence from the typical quantization error minimization objective renders the TDQ scheme less effective in this particular context.
>
> These findings collectively affirm that TDQ exhibits broad compatibility with a variety of QAT methods, thereby reinforcing its potential as a versatile tool in the quantization domain. We will include these supplementary results and an analysis of the observations in the camera-ready version of our paper.
>
> **W5: thorough proof reading**
>
> Thank you for the constructive feedback. we will conduct sincere proof reading and refine the writing.
>
> ---
>
> ### Limitations
>
> **L1: Input-dependent quantization**
>
> As detailed in our response to weakness 1, extracting instance-wise dynamic information incurs significant latency costs. Furthermore, the instance-wise quantization interval poses challenges by preventing batched references. Nonetheless, in specific hardware environments where the batch size is 1, or when the costs of extracting statistical information can be amortized, utilizing such information alongside time data may lead to performance improvements. We see this integration as an exciting avenue for future research, as it could offer additional enhancements in efficiency and effectiveness.
>
> ---
> ### References
> [1] Vasu, Pavan Kumar Anasosalu, et al. "MobileOne: An Improved One Millisecond Mobile Backbone." Proceedings of the IEEE/CVF Conference on Computer Vision and Pattern Recognition. 2023.

---

> > ### Comment · Reviewer_xsTz · 2023-08-16
> >
> > I thank the authors for the rebuttal.
> > One question still remains regarding W3. The claims regarding time dependency behaviours seem to be based on hand-picked observations. What is the exact ratio of layers that have time dependency behaviours in comparison to layers that do not have time dependency similar to `mid.attn_1.proj_out.0`? and also which layers tend to have time dependency behaviours and which layers tend to lack time dependency?
> >
> > Best,
> > Reviewer xsTz

---

> > > ### Author Response · Authors · 2023-08-21
> > > **Exact ratio of layers that have temproal depenence**
> > >
> > > Thank you for your detailed feedback. It is important to quantify this ratio to support our claim that temporal dependence exists in diffusion models.
> > > To measure the exact ratio of layers with high temporal dependence, we calculated the *correlation* between time steps and the activation range of each time step. Specifically, we computed the absolute value of the Pearson correlation coefficient and nlcor[1], as some layers may exhibit non-linear temporal dependence. Both metrics produce values between 0 and 1, with higher values indicating stronger temporal dependence.
> > >
> > > |method | r>0.3 | r>0.5 | r>0.7 |
> > > | --- | --- | --- | --- |
> > > | absolute pearson | 71.6% | 62.1% | 38.8% |
> > > | nlcor[1] | 72.8%| 47.5% | 31.0% |
> > >
> > >
> > >
> > > The table reveals that about 60% of the layers exhibit moderate temporal dependence (r > 0.5), and roughly 35% exhibit strong temporal dependence (r > 0.7). This implies that many layers in diffusion models indeed possess strong temporal dependence. Therefore, we can deduce that employing the TDQ module for these layers is important. Additionally, we can assert that these metrics serve as proper representations of temporal dependence, supported by values `r=0.19` for `mid.attn_1.proj_out.0` and `r=0.90` for `down.1.atton.0.proj_out.0` in Figure 7 of the Supplementary Materials.
> > >
> > > ---
> > > Reference
> > > [1] Ranjan, Chitta, and Vahab Najari. “nlcor: Compute Nonlinear Correlations.” Research Gate (2020) (2020).

---

### Official Review · Reviewer_7wR5 · 2023-07-06

**Soundness:** 2 fair
**Presentation:** 3 good
**Contribution:** 2 fair
**Rating:** 5
**Confidence:** 5

**Summary:**

This paper proposes Temporal Dynamic Quantization (TDQ), a new quantization approach for diffusion models. This method utilizes a learned module to adjust the quantization interval with respect to timestep to deal with activation quantization errors. Unlike conventional dynamic quantization techniques, the TDQ approach has no computational overhead during inference and is compatible with both post-training quantization (PTQ) and quantization-aware training (QAT). Experiments have been performed using DDIM on the CIFAR-10 dataset and LDM on the LSUN-Churches dataset with TDQ and various traditional QAT and PTQ approaches.

**Strengths:**

- The paper is well organized and written with clear motivation; the idea of addressing the rapid changes in input activation to reduce the truncation & rounding errors with a time-dependent quantization configuration is natural and interesting.
- The proposed TDQ framework can be applied to both QAT and PTQ cases.
- Experimental results demonstrate that the proposed framework can outperform several previous QAT and PTQ methods. Ablation studies have been conducted to show that TDQ can generalize to scenarios with fewer denoising steps and analyze output dynamics across layers.

**Weaknesses:**

My concerns are mainly about the evaluation in this paper:
- The proposed TDQ framework primarily aims to deal with the activation quantization, but it is not specified whether the act-to-act matrix multiplications (attn and v) in the attention layers are quantized. By examining the code attached in the supplementary materials, it seems like those computations still remain in FP. However, they can take a significant portion of the compute & memory overheads and are fully quantized in closely related work [1], while past literature [1, 2] has pointed out that the post-softmax attention scores are usually hard to quantize (e.g. PTQ4DM suffers from performance degradation after quantizing them according to Appendix E in [1]). It is unclear if TDQ can deal with this performance degradation without providing experimental results with act-to-act matmuls in attention layers quantized.
- It is not accurate to call PTQ4DM the 'state-of-the-art study'—Q-Diffusion [1] was proposed around the same time as PTQ4DM and contained results that went down to 4-bit weights on both CIFAR and LSUN datasets, and achieved superior results on CIFAR. Its code (with quantized checkpoints) was released in April. I would highly recommend the authors include and compare Q-Diffusion results for CIFAR and LSUN-Churches experiments as it is a highly related work that natively covers many experimental settings in this work that are not in PTQ4DM.

[1] Li, X., Liu, Y., Lian, L., Yang, H., Dong, Z., Kang, D., ... & Keutzer, K. (2023). Q-diffusion: Quantizing diffusion models. arXiv preprint arXiv:2302.04304.
[2] Yuan, Z., Xue, C., Chen, Y., Wu, Q., & Sun, G. (2021). PTQ4ViT: Post-Training Quantization Framework for Vision Transformers with Twin Uniform Quantization. arXiv preprint arXiv:2111.12293.

**Questions:**

- Can you further explain the output dynamics in Figure 9 and Appendix Figure 6? E.g. what do blue and red curves stand for and how to interpret them?
- Looking at Appendix Table 1, I am curious if some simple baselines following the same principle can be helpful, such as using multiple quantizers at different time steps under the LSQ setting e.g. using a separate activation quantizer every 20 steps.
- How sensitive is the TDQ module (MLP) to the hyperparameters? Are there different design choices for the training process that cause different results? For instance, it is mentioned in the paper that initialization is crucial – are there experiments showing that a different initialization (e.g. gaussian, Xavier, zero…) can make results worse?

**Limitations:**

The authors adequately addressed all limitations.

---

> ### Author Rebuttal · Authors · 2023-08-09
>
>
> Thank you for bringing up important points with productive feedback.
>
> ---
>
> ### Weaknesses
>
> **W1: Quantization Configuration Details for Different Layers**
>
> Thank you for raising the pivotal aspect of activation quantization for attention multiplicands. To clarify, our implementation does indeed account for the quantization of the Query, Key, and Value activations. In scenarios involving matrix multiplication between the Query and Key, the full benefits of low-precision multiplication are harnessed.
>
> However, you rightly identified a nuance: the attention scores following softmax were not subjected to quantization. While Q-Diffusion only experimented with reducing the activation bit width to 8 bits, our experiments reduced it down to 3 bits. In this low-bit configuration, we found that excessive quantization of the attention scores led to a substantial decline in output quality.
>
> To manage this, we refrained from applying quantization to the attention score, considering its high sensitivity to the quantization process. It's noteworthy to mention that this approach was consistently applied; none of the baselines in our experiment incorporated quantization in this segment either. Although specialized quantization techniques exist for post-softmax scores, we did not employ them, as this aspect was not the primary focus of our experiments.
>
> In response to concerns regarding the application of quantization to the softmax score, we conducted a supplementary experiment. We wanted to validate whether TDQ performs well, even in scenarios where the output of attention score is quantized. The experiment was designed with four distinct scenarios in mind, and the LSUN-church 256 dataset with QAT was utilized.
>
> The results are as follows:
>
> 1. **Without softmax quantization:**
>     - LSQ: FID score of 5.06
>     - TDQ: FID score of **4.64**
> 2. **With softmax Quantization:**
>     - LSQ: FID score of 5.21
>     - TDQ: FID score of **4.99**
>
> These findings quantitatively affirm that TDQ consistently achieves superior quality results, even when attention quantization is introduced. In the forthcoming camera-ready version, we will provide more extensive details and additional results related to the quantization of the attention score.
>
> **W2: Comparsion with Q-Diffusion**
>
> We have thoroughly prepared a response to address your concern. Kindly refer to the global response section where we detail the comparison with Q-Diffusion.
>
> ---
>
> ### Questions
>
> **Q1:  Interpretation of Fig. 9 in main paper and Fig 6 in Appendix .**
>
> In our visualizations, the blue curve represents the quantization interval for each time step, while the red curve illustrates the fluctuations in the magnitudes of the activations over each time step. As expected, the ideal behavior is for the quantization interval (blue curve) to expand as the variance (depicted by the red curve) amplifies. This expansion ensures that quantization errors are minimized.
>
> For many layers, this desired behavior is evident. For instance, the layer `Input_blocks.1.0.emb_layers.1.act_func`, as showcased in Appendix Figure 6, clearly demonstrates this pattern. However, certain layers, such as `Middle_block.0.out_layers.3.act_func` depicted in the same figure, don't exhibit a pronounced effect from TDQ. This discrepancy arises primarily due to the combination of significant variations in data and the layer's low temporal dependency. In scenarios like these, the output from TDQ tends to stabilize, converging to a specific value. This behavior mirrors that of LSQ, where the quantization interval becomes essentially constant.
>
> **Q2. Learning quantization interval directly for each time step**
>
> We have thoroughly prepared a response to address your concern. Kindly refer to the global response section where we detail the comparison with the direct interval learning for each time step.
>
> **Q3. Robustness of TDQ module**
>
> In our paper, when we refer to the sensitivity of initialization, we are specifically addressing the initial value of the quantization interval. It's essential that the quantization interval align with the data distribution right from random initialization. Otherwise, a too-small interval might truncate valuable information, while a too-large one could increase discretization errors significantly. To ensure stable convergence, we initialize the bias of the last layer of the TDQ module following the initialization method described in the LSQ paper.
>
> Additionally, the TDQ module seems relatively insensitive to other design choices. For example, it exhibits comparable accuracy across different random initialization methods, as evidenced by the table below (except for Gaussian initialization). In some instances, TDQ fails to converge (though it's worth noting that TDQ enhances convergence stability, and existing quantization schemes like LSQ are more prone to failure). However, once trained properly, the output quality is consistently good.
>
> - **TDQ Weight Initialization ablation**
>
>
>     | (FID↓) | He (Default) | Gaussian | Zero | Xavier |
>     | --- | --- | --- | --- | --- |
>     | CIFAR10 W4A4 | **4.80** | Fail | 4.98 | 4.90 |
>
> The effect of varying the number of layers was also examined, and the results remained consistent across different configurations. We empirically selected four layers as our default number. These findings and a more detailed explanation will be included in the supplementary material of the camera-ready paper.
>
>
> - **TDQ Number of layer ablation**
>
>
>     | (FID↓) | 2 Layer  | 4 Layer (Default) | 8 Layer  |
>     | --- | --- | --- | --- |
>     | CIFAR10  W4A4 | 5.18 | **4.80** | 4.88 |

---

> > ### Comment · Reviewer_7wR5 · 2023-08-19
> > **Response to Authors**
> >
> > Dear Authors,
> >
> > Thank you for the comprehensive rebuttal and the clarifications provided. The additional results have mostly addressed my concerns. I would be inclined to raise my scores if those experiments and discussions (e.g. comparison with Q-Diffusion, learning quantization interval directly for each time step etc.) will be included in the final version of the paper.
> >
> > Best regards,
> > Reviewer 7wR5

---

> > > ### Author Response · Authors · 2023-08-21
> > > **Response to Reviewer**
> > >
> > > Dear reviewer 7wR5,
> > >
> > > We will gladly include these experimental results and discussions in the final version of our paper. Thank you very much for taking the time to review our work and provide valuable feedback. We believe that these experiments and discussions have undoubtedly enhanced the quality of the paper.

---

### Official Review · Reviewer_qrrn · 2023-07-06

**Soundness:** 3 good
**Presentation:** 2 fair
**Contribution:** 2 fair
**Rating:** 4
**Confidence:** 3

**Summary:**

The authors present and empirically evaluate a novel quantization scheme for diffusion models. Static quantization of the evolving activation distribution in diffusion models leads to performance degradation. By using a quantization range which is a function of the timestep, the authors improve performance.

**Strengths:**

The paper is very well structured and easy to follow and provides clear and concise descriptions of both introductory quantization literature, as well as diffusion model literature. The problem definition and proposed solution are also very well described, especially in Figures 2 and 3. Empirically, evaluation under multiple metrics (FID and ID) is always appreciated. The experimental setup is well described.

**Weaknesses:**

- The form of the TDQ module (network size, parameters, structure) isn't provided.
- Not a particularly major issue but there are a few spelling/grammatical mistakes. For brevity I won't list all the errors but online tools like Grammarly are free to use and would help in this regard.
- For other weaknesses, please see my questions.

**Questions:**

- This paper seems slightly contradictory. In section 3.3, the authors assume that the time evolution of the activation distribution is constant, which permits the use of their TDQ module. The authors also make note of the high cost of inference (clearly outweighing the training costs). So why is the TDQ network module neccessary? Why not just learn the the activation distribution directly for each timestep. Learning this function through a network (rather than simply per timestep) would impose a smoothness contraint on the function. Is this the only benefit?
- Have the authors provided an ablation study where the activation variations, as in Figure 9, are used rather than the TDQ module? Again, why not just compute these beforehand? Does using this pre-computed activation variance perform better? Perhaps some more detailed ablations studies are required?
- How robust is this TDQ module to training? No error bars are provided so it's unclear if there are any failure cases, or if the predicted intervals remain quite consistent across trials. Given the assumption in section 3.3, they should always converge to the same interval values. Do the authors observe this?

**Limitations:**

- The limitations of this work aren't explicitly outlined. For example, whilst the authors show a couple of plots illustrating the time evolution of the activation distribution, the assumption doesn't appear to be investigated. What are the effects of this assumption?
- In addition, it doesn't appear that the authors have outlined any limitations wrt to how text conditioning (v. common for diffusion models in practice) could break this assumption. Perhaps useful for future work, but the authors could make a comment for the sake of practitioners looking to employ this method?
- I see no wider negative societal implications of this work.

---

> ### Author Rebuttal · Authors · 2023-08-09
>
> Thank you for your valuable feedback and constructive reviews.
>
> ---
>
> ### Weakness
>
> **W1: TDQ Module Specifications**
>
> In this manuscript, we present the TDQ module, a carefully designed 4-layer MLP with ReLU activation. The module takes 128 input elements and has a hidden layer with 64 elements. We will add this details in camera-ready iteration.
>
> **W2: Addressing Spelling and Grammar Mistakes**
>
> We appreciate your thorough examination. We will review the grammar and wording once more, utilizing professional tools.
>
> ---
>
> ### Questions
>
> **Q1: The necessacity of TDQ module**
>
> During inference, the TDQ Module's computational cost matches that of learning activation distribution directly for each timestep. The learnable parameters of the TDQ Module undergo training during the PTQ/QAT process, and once this process concludes, these parameter values become static. Because its input depends solely on the time step, the quantization interval to each timestep can be pre-calculated offline. Thus, the TDQ Module exhibits no shortcomings when compared to the direct learning of activation distribution for each timestep.
>
> Moreover, the TDQ Module presents clear advantages over the method of per-timestep learning. Firstly, by evaluating activation alterations collectively across multiple timesteps, the learning process tends to be more stabilized, a point which the reviewer rightly highlighted. Furthermore, even if there are discrepancies in the length of the timestep during training and inference (e.g., SDE[Song et al.] uses 1000 timestep during training but employs 3000 for sampling), the TDQ Module can determine the apt quantization interval for the generation step, referencing the parameters it has learned.
>
> To address the reviewer's feedback and further validate our assertions, we've carried out additional experiments. Detailed results can be found in the global response 1. We kindly request you to refer to that section for more insights.
>
> **Q2: Ablation Study on the Use of Activation Variation Versus the TDQ Module**
>
> Prior work in QAT (e.g., PACT, QIL, LSQ) and PTQ (e.g., AdaRound, BRECQ) has emphasized the importance of determining the ideal quantization interval. This is crucial in balancing between truncation and rounding errors. These methods typically treat the quantization interval as a learnable parameter, and refine it through gradient descent to minimize objective function like task loss or reconstruction error. Inspired by these principles, the TDQ module is designed to identify optimal quantization ranges for each timestep using gradient descent based on established PTQ/QAT techniques.
>
> To verify our method's efficiency, we contrasted performance of TDQ-learned temporal intervals with pre-computed maximum intervals. In CIFAR-10 DDIM W4A4 QAT, TDQ yielded an FID score of **4.80**, while the pre-computed maximum's score of 6.30. This highlights TDQ's superior output quality over precomputed statistics based method. A comprehensive ablation study will be elucidated in our finalized version.
>
> **Q3: Robustness of the TDQ Module**
>
> To evaluate the robustness of the TDQ module, we perform three trials with distinct seeds for both DDIM CIFAR10 QAT and LDM churches QAT. Please check *Table 1* at our Global Response PDF.
>
> As shown in Table, TDQ consistently outperform of conventional QAT methods, indicating its enhanced stability over LSQ.  The limited adaptability of LSQ—relying on a single quantization interval across all time steps—can occasionally lead to suboptimal performance. In contrast, TDQ dynamically updates the interval at every time step, taking into account adjacent distributions. We will refresh our paper's tables, incorporating the mean variance observed across multiple trials.
>
> Furthermore, an intriguing observation across various trials is the convergence of quantization intervals. In disparate trials, these intervals tended to approach nearly identical values at each time step. This consistency further underlines TDQ's robustness. For a visual representation, please kindly refer to the  *Figure 2* at our Global Response PDF.
>
> ---
>
> ## Limitations
>
> **L1: The Time Evolution of the Activation Distribution**
>
> A limitation of our assumption resides in the potential non-marginal time-dependent divergence of the activation distribution in specific models or layers.
>
> For example, the LDM model manifests a more subtle temporal variation compared to a pixel-wise diffusion model, resulting in only a modest performance enhancement when employing the TDQ module. Similarly, even within the DDIM models, the deeper layers may exhibit less temporal dependency, a trait that might appear counter intuitive.
>
> However, it is essential to recognize that even in such instances, the TDQ module guarantees convergence, yielding consistent output across time steps. This alignment ensures that the output quality is compatible with that of existing PTQ/QAT algorithms, thus preserving the module's value even where the time-related divergence might be less significant.
>
> **L2: Effect of TDQ Module for Text Conditioning**
>
> Thank you for raising an important issue concerning the effect of text conditioning. We extracted 12 images for three distinct texts from Stable-Diffusion and depicted their activation distributions in *Figure 1* of the Global Response PDF.
>
> As revealed by the figure, input distributions display minor variations depending on text conditions. However, the feature map's overall trends seem to be more reliant on time than the specific text condition. Thus, we have found that our method can be applied to text-conditioned scenarios such as Stable-Diffusion without hindrance.
>
> The extension of the TDQ Module to cater to text conditioning presents an intriguing aspect and points to a promising avenue for future research. To enhance understanding and offer a visual perspective, we'll include the mentioned figure and a fitting explanation in the supplementary material.

---

> > ### Comment · Reviewer_qrrn · 2023-08-17
> > **Response to Rebuttal**
> >
> > Dear Authors,
> >
> > First, I would like to thank you for your detailed rebuttal and additional experiments pertaining to the comments made. Whilst the response goes some way to answer the questions raised, I would like to clarify a few outstanding concerns:
> >
> > Q1: Whilst the TDQ module outperforms the new baselines on most tasks, I am curious about other factors involved. For example, what was the training setup? Was any regularisation across time implemented (as mentioned in my initial question) to impose the smoothness of the changes in the activation distribution. Did the authors find this simpler/easier to implement than the TDQ module?
> >
> > L2: Whilst I agree that the general trend stays somewhat similar with text conditioning, there are certainly subtle differences in the time evolution. Does this affect performance? I think it would be useful to present some more concrete performance metrics, especially across multiple models and more text prompts. That being said, it could perhaps make for very interesting future work.
> >
> > I would welcome any comments by the authors, especially with regards to Q1, which seems to be quite crucial to the proposed contribution.
> >
> > Many thanks.

---

> > > ### Author Response · Authors · 2023-08-21
> > > **Response to Reviewer qrnn**
> > >
> > > Dear reviewer qrnn,
> > > Thank you for bringing up important points.
> > > ### Q1
> > >
> > > - **Training Setup**: We wish to emphasize that our training for TDQ did not use a specialized setup. We adopted the same optimizer, learning rate, scheduler, and weight decay as used for the baseline training. The TDQ module can be integrated into any static quantization activation algorithm without modifications to the existing training pipeline. For transparency, we have provided the source code, enabling easy reproduction of our results without special setups for the TDQ module.
> > >
> > > - **Smoothness Regularize**: No dedicated regularizer was employed to govern the smoothness of the activation distribution. As mentioned earlier, we adhered to the existing training pipeline without modifications. However, it is worth noting that TDQ can potentially act as an implicit smoothness regularizer due to the inherent characteristics of Multilayer Perceptrons. These networks naturally yield similar outputs for analogous inputs, consequently leading to a trend where the truncation of activations at similar time steps tends to be the same.
> > >
> > > - **Simpler Implementation**: We respectfully assert that the TDQ module we've proposed presents the most straightforward approach to addressing the temporal variation of activation distribution. As shown in Table 2 of the global response, when the TDQ dimension is adjusted for the target task's complexity, it significantly outperforms S_{N}. The TDQ module is user-friendly and integrates smoothly into existing frameworks. While there may be a need to adjust certain internal hyperparameters like the hidden dimension for specific tasks, we have provided a comprehensive implementation guideline that is applicable across various datasets. This makes it easier for researchers and developers to harness the advantages of TDQ without extra effort.
> > >
> > > ### L2
> > > Thank you for your insightful feedback, which has certainly enhanced the quality of our work. We concur with the suggestion to explore the proposed technique for text conditioning in future work. To offer a clearer evaluation in response to your comments, we carried out further experiments using QAT for the text-conditioning diffusion model. However, due to the constraints of the rebuttal period, we regret that our results may appear somewhat preliminary. To measure the performance of TDQ Module for text conditioning, we applied our QAT-TDQ to Stable Diffusion (V1.4). We employed the CLIP score as our evaluation metric, a common metric for generative models, particularly for reference-free models like Stable Diffusion.Specifically, the LAION-Aesthetics(6.5) dataset was employed for QAT. In the evaluation of the clip_score, we randomly selected 15 texts from the MS-COCO dataset and produced 1000 images for each text. Subsequently, we computed the clip_score for both the Full-Precision (FP) Stable Diffusion model and our quantized TDQ version.
> > >
> > > | Text idx | 1     | 2     | 3     | 4     | 5     | 6     | 7     | 8     | 9     | 10     | 11     | 12     | 13     | 14     | 15     |
> > > |----------|------------|------------|------------|------------|------------|------------|------------|------------|------------|------------|------------|------------|------------|------------|------------|
> > > | FP           | 0.315 | 0.314 | 0.310 | 0.302 | 0.311 | 0.315 | 0.316 | 0.30 | 0.320 | 0.317 | 0.329 | 0.312 | 0.348 | 0.282 | 0.347 |
> > > | TDQ_W6A6   | 0.299 | 0.309 | 0.304 | 0.291 | 0.300 | 0.305 | 0.317 | 0.290 | 0.297 | 0.294 | 0.309 | 0.292 | 0.339 | 0.276 | 0.332 |
> > >
> > > As shown in the table, the performance degradation for each text is not substantial, delivering a consistent performance drop of around 0.01 to 0.02. These results clearly demonstrate that TDQ remains minimally affected by subtle distribution variations prompted by text conditioning, making it well-suited for application in text-to-image models like Stable Diffusion or Imagen.
> > > Nevertheless, we posit that incorporating text conditioning information has the potential to enhance the quality of the output. While this might increase the overhead of step size generation owing to text-dependent features, such an increase is likely to be marginal compared to input-dependent quantization, and could be comparable to the overhead associated with the TDQ module. We view this avenue as a promising extension for future work. Once again, we are genuinely grateful for your insightful feedback.

---

### Author Rebuttal · Authors · 2023-08-09

# Global response for reviewers
We appreciate all the reviewers for taking time to review our work and giving valuable comments and feedbacks that help to improve the completion of our paper.

---
## 1. Learning quantization interval directly for each time step

We highlighted that as the activation distribution evolves over time, the quantization interval should adapt correspondingly. Consequently, we introduced the TDQ Module. However, several reviewers inquired if a more direct approach might be viable: namely, could the quantization interval be trained using scalar variables assigned to each time step, sidestepping the need for more intricate structures like the TDQ Module? To address this, we carried out experiments contrasting the performance of the TDQ Module with a configuration where the quantization interval parameter is directly learned. Furthermore, we introduced an alternative concept: employing a shared quantization interval across multiple time steps. In the table we provide, the notation \( S_{N} \) signifies that there are a total of \( N \) learnable quantization interval parameters that cover 1000 / N time steps uniformly.

- **Table 1. Quantization-Aware Training (QAT) Results**


    | (FID↓) | TDQ | S_1 (LSQ) | S_10 | S_50 | S_100 | S_1000 |
    | --- | --- | --- | --- | --- | --- | --- |
    | CIFAR10-DDIM W4A4 | **4.48** | 7.3 | 4.75 | 5.17 | 4.88 | 5.53 |
    | CIFAR10-DDIM W3A3 | **6.48** | 7.63 | 8.89 | 6.92 | 6.82 | 9.94 |
    | Church-LDM W4A4 | **4.64** | 5.06 | 5.17 | 5.07 | 4.78 | 5.10 |
    | Church-LDM W3A3 | 6.57 | 7.21 | 6.70 | **6.47** | 6.64 | 6.86 |
- **Table 2. Post-Training Quantization (PTQ) Results**

    | (FID↓) | TDQ | TDQ_Thin | S_1(LSQ) | S_2 | S_4 | S_10 | S_20 |
    | --- | --- | --- | --- | --- | --- | --- | --- |
    | Church-LDM W4A8 | 44.88 | **28.74** | 76.36  | N/A | 33.19 | 37.63 | 29.87 |
    | Church-LDM W4A6 | 120.53 | 55.27 | 158.07 | 193.11 | 185.19 | 72.37 | **47.39** |
    | ImageNet-LDM W4A6 | 41.23 | **16.96** | 47.26 | - | 85.11 | 21.71 | 17.38 |

In our QAT experiments, TDQ exhibited outstanding or near-optimal performance, while configurations with \( S_{50} \) or \( S_{100} \) also demonstrated significant effectiveness. The TDQ module's success lies in its ability to enable continuous and stable learning of the quantization interval, taking into consideration the evolution of the activation distribution over neighboring time steps. This results in generator output that is more apt for generation.

Simultaneously, the concept of a shared quantization interval across multiple time steps can partially capture the benefits of TDQ, significantly enhancing output quality over the \( S_{1} \) configuration. However, TDQ still achieves superior output quality and offers greater versatility, particularly in scenarios such as employing different timesteps during training and inference.

In our PTQ experiments, while the standard TDQ module yielded a notable performance boost over \( S_1 \), it lagged behind configurations such as \( S_{20} \). Our investigations revealed that the default TDQ module was prone to overfitting due to two primary reasons: 1. Training the MLP with a limited calibration dataset (containing only 256 samples) proves challenging. 2. The relatively brief training iteration, typically 5000 in BRCEQ, finds it hard to filter out the high-frequency component present in frequency-encoded input features. To mitigate these constraints, we introduced a streamlined version of TDQ—referred to as **TDQ Thin**. This refined module uses a 3-layer MLP with a mere 16 hidden dimensions and omits the frequency encoding for time steps. Notably, this adaptation showcased results that either surpassed or matched the best performances observed in the \( S_{N} \) experiments.

We deeply appreciate the constructive feedback. It has provided insights enabling us to refine TDQ to be more PTQ-friendly. We are committed to incorporating these insights judiciously, ensuring they feature prominently in the finalized version of the paper.

---

## 2. Comparison with Q-Diffusion

It's important to highlight that at the time of our paper's composition, Q-Diffusion was not a peer-reviewed paper and was only available on Arxiv. Although it is currently marked as an ICCV2023 accepted paper on their GitHub, no official accepted list has been published. Moreover, Q-Diffusion only provided inference code, lacking the essential training (calibration) code. Our attempts to reproduce their results during the submission period were met with challenges. Fortunately, the recently released paper PTQD[1] offered a codebase that reimplemented Q-Diffusion. We utilized this codebase to conduct our experiments. However, it's worth noting that PTQD's implementation differs slightly from the original Q-Diffusion paper, such as the absence of shortcut-splitting quantization and its use of symmetric per-tensor weight quantization instead of asymmetric per-channel weight quantization found in Q-Diffusion.
The comparative results are tabled below:

- **Table 3. Performance comparison with Q-Diffusion**
    | (FID↓) | Church-LDM W4A8 | Church-LDM W4A6 | ImageNet-LDM W4A6 |
    | --- | --- | --- | --- |
    | QDiffusion  | 76.36 | 158.07 | 47.26 |
    | + TDQ | **28.74** | **55.27** | **16.96** |


As demonstrated in the table, our TDQ method shows significant improvement compared to Q-Diffusion. These results underscore the importance of utilizing different quantization intervals for different time steps when quantizing diffusion models. Because TDQ and QDiffusion are orthogonal methods, the integration of TDQ over QDiffusion offsers notable performance improvement.  We plan to include these results (or an updated version if the official code becomes available) in the camera-ready version of our work.


---
### References

[1] He, Yefei, et al. "PTQD: Accurate Post-Training Quantization for Diffusion Models." arXiv. (2023).

---

### Decision · Program_Chairs · 2023-09-21

**Decision:**

Accept (poster)

**Comment:**

Most reviewers are positive about the paper. The reviewer with borderline rejection has some concerns, but the rebuttal has provided an explanation. This work has presented an interesting quantization method that dynamically adjusts quantization intervals, and the proposed method is compatible with existing QAT and PTQ algorithms. The experiments are thorough and convincing. The authors may also add the comparison to Q-Diffusion in this camera-ready version. After discussing this paper with SAC, I would recommend acceptance.